# Neural circuits underlying context-dependent competition between defensive actions in *Drosophila* larvae

Maxime Lehman[1], Chloé Barré[2,3,6], Md Amit Hasan[1,6], Benjamin Flament[1], Sandra Autran[1], Neena Dhiman [4,5], Peter Soba[4,5], Jean-Baptiste Masson [2,3] & Tihana Jovanic [1] ✉

To ensure their survival, animals must be able to respond adaptively to threats within their environment. However, the precise neural circuit mechanisms that underlie flexible defensive behaviors remain poorly understood. Using neuronal manipulations, machine learning-based behavioral detection, electron microscopy (EM) connectomics and calcium imaging in *Drosophila* larvae, we map second-order interneurons that are differentially involved in the competition between defensive actions in response to competing aversive cues. We find that mechanosensory stimulation inhibits escape behaviors in favor of startle behaviors by influencing the activity of escape-promoting second-order interneurons. Stronger activation of those neurons inhibits startle-like behaviors. This suggests that competition between startle and escape behaviors occurs at the level of second-order interneurons. Finally, we identify a pair of descending neurons that promote startle behaviors and could modulate the escape sequence. Taken together, these results characterize the pathways involved in startle and escape competition, which is modulated by the sensory context.

The ability to avoid danger and threats is essential for animal survival[1–4]. The defensive strategies used by animals to evade potential dangers often include various modes of locomotion to escape from the perceived threat or to freeze and thus impede detection by the aggressor. Additionally, animals may execute protective actions designed to minimize the exposure of sensitive areas of their bodies[5]. The type of behavior that is performed depends on the type of danger as well as on the specific context in which the danger is encountered.

While some defensive behaviors are stereotyped to ensure rapid responses to threatening stimuli, the behaviors also need to be flexible enough to permit animals to respond differently to threats depending on the environmental context and their internal needs[5–8]. Escape behaviors range from very simple reflex-like actions[9–11] to more complex actions requiring cognitive processes that rely on memory and decision-making[5,12,13]. Animals must first decide whether to respond and then how to respond to threats. Both decisions could be affected by the type and degree of the threat, the external context, or the internal or behavioral state. For example, in response to looming stimuli, hungry crayfish freeze rather than perform a tail flip[14], and feeding leeches ignore mechanical stimuli by blocking the transmission of mechanosensory information to central circuits through presynaptic inhibition of mechanosensory terminals[15]. In mice, the decision of whether to freeze or escape to shelter depends on the environmental context and the availability of a shelter in a memorized

[1]Université Paris-Saclay, CNRS, Institut des neurosciences Paris-Saclay, 91400 Saclay, France. [2]Institut Pasteur, Université Paris Cité, IHU reConnect, IHU ICE, CNRS UMR 3571, Decision and Bayesian Computation, 75015 Paris, France. [3]Epiméthée, INRIA, 75013 Paris, France. [4]Institute of Physiology and Pathophysiology, Friedrich-Alexander-Universität Erlangen-Nürnberg, 91054 Erlangen, Germany. [5]LIMES Institute, Department of Molecular Brain Physiology and Behavior, University of Bonn, Carl-Troll-Str. 31, 53115 Bonn, Germany. [6]These authors contributed equally: Chloé Barré, Md Amit Hasan. ✉e-mail: tihana.jovanic@cnrs.fr

location[16]. Moreover, defensive behaviors are often not single actions but multiple actions that need to be organized in sequence[5,6,17–19], the order of which may also vary depending on the context.

The neural circuit mechanisms underlying the flexibility of defensive behaviors, the selection between various competing actions, and their sequential organization on the basis of context remain unclear. In this study, we address this question in *Drosophila melanogaster* larvae, which, due to their genetic tractability and the availability of the electron microscopy (EM) connectome[18] of its 10,000-neuron central nervous system (CNS), allows the mapping of circuits at the synaptic and cellular levels. The ease of circuit mapping, combined with the rapid life cycle and the ease of automated and quantitative behavioral approaches, makes it an excellent model for relating neural circuit structure and function underlying defensive behaviors[20].

To avoid and escape various threats, larvae perform different actions and use different strategies depending on the nature and degree of threat. For example, we have previously shown that an air puff, an aversive stimulus, can trigger five different types of actions: two startle-like actions, Hunch and Stop, and three escape actions, Back-up, Crawl, and Bend[19]. These actions can be organized into probabilistic sequences[17,19]. In response to nociceptive stimuli, however, larvae exhibit escape behaviors consisting of C-shapes and Rolls[18,21–23].

In this study, we identified second-order interneurons that are located downstream of the previously described startle and escape circuit[17,19] and that underlie the inhibition of startle responses while simultaneously promoting an escape sequence. We further found that the sensory context modulates the escape sequence induced by optogenetic activation of these interneurons. Modulation of the behavioral output depends on the strength of activation of the second-order interneurons relative to the strength of the mechanosensory context introduced by the application of an air puff. Taken together, our work reveals that the level of activation of the identified neurons can gate specific responses depending on the context. These neurons thus contribute to circuit computation for selecting appropriate defensive behavior, specifically in the presence of mechanosensory and nociceptive aversive sensory cues.

## Results

### EM connectivity analysis reveals candidate second-order interneurons for avoidance responses to a mechanical stimulus

We have previously characterized the behavioral response of larvae to the mechanical stimulus of an air puff. We found that, in response to air puff stimulation, larvae performed a probabilistic sequence of five different and mutually exclusive avoidance actions—Hunch, Bend, Back-up, Stop, and Crawl—and identified the pathways in the mechanosensory network that underlie these behaviors[19] (Fig. 1a). The circuit for the selection between the two most prominent actions in the sequence—Hunching (head-retraction) and Bending—has been previously described in detail[17]. The circuit is composed of chordotonal mechanosensory neurons that sense the air puff, a layer of different types of inhibitory neurons that implement competitive interactions, and a layer of two projection neurons: Basin-1 and Basin-2, which receive inputs from chordotonal sensory neurons. Inactivation experiments and the model of that circuit predict that the coactivation of both projection neurons gives rise to a Bend, whereas the activation of Basin-1 results in Hunching (Fig. 1b). These observations suggest that the activation of Basin-2 effectively inhibits Hunch. To determine how the information is decoded downstream of Basin neurons to give rise to Hunching or Bending, we examined the EM connectome of neurons downstream of Basin-1 and Basin-2, which could be involved in these behaviors. One such candidate neuron is A19c, previously identified in a behavioral screen labeled by the genetic driver R11A07[19]. R11A07 also labels a thoracic neuron with a descending projection). In that screen, driving tetanus toxin (TNT) with the R11A07 driver

resulted in an increase in the Hunching probability in response to air puff, suggesting that A19c inhibits Hunch. Interestingly, A19c receives inputs from Basin-2 and another Basin neuron type, Basin-4 neurons[18,19] (Fig. 1c, d; Supplementary Fig. 1a–e; Source Data 1). Basin-4 neurons, similar to Basin-2 neurons, also inhibit the Hunch response[17].

Another group of candidate neurons downstream of Basins that could be involved in air puff responses are the interneurons A08m and A08x, which receive the largest fraction of inputs from Basin-1 and Basin-3 and a relatively smaller percentage of inputs from Basins 2 and 4 (Fig. 1c, e, f, Supplementary Fig. 1f–m, Source Data 2). A08m receives slightly more inputs from Basin-3 than Basin-1 in the first abdominal segment (Fig. 1e, Supplementary Fig. 1f–i, Source Data 2). A08x receives most of its Basin inputs from Basin-1 (in all segments where Basins were reconstructed) (Fig. 1f, Supplementary Fig. 1j–m). A08x and A08m neurons are part of a group of five neurons called early-born Even-skipped expressing lateral neurons (ELs)[24,25]. These neurons respond to mechanical stimulation, and their optogenetic activation has been shown to induce escape Rolling[25].

### Second-order interneurons in a mechanosensory network are differentially involved in startle and escape responses to air puff

To determine whether A19c and A08m,x could be involved in decoding Basin-1 and Basin-2 activities to induce appropriate motor outputs, we investigated the role of A19c and early-born ELs in air-puff-induced sensorimotor decisions. For this purpose, we used high-throughput behavioral assays that we have previously established[17,19,26] and an updated classification method that we developed in this study. The new classifiers, in addition to detecting Hunch, Stop, Back-up, Crawl, and Roll, discriminate two types of Bend behaviors: Head Casting and Static Bending (Fig. 1a; see Methods for details). Head Casting consists of the larva swiping its head to explore the environment once or multiple times. It occurs both in the presence and absence of sensory stimulation in the context of, for example, navigating sensory gradients or foraging, respectively[27–35]. Static Bending is a protective action that is triggered by stimulation. Using the updated classification method, we confirmed that the inactivation of R11A07 neurons by driving TNT resulted in an increased Hunching probability in response to air puff (Supplementary Fig. 2a–c)[19]. A similar phenotype was observed when optogenetic inactivation was performed using GtACR1 and red light for stimulation (Supplementary Fig. 2d–f). When the green light was used for optogenetic stimulation, less Head Casting, more Stopping and more Backing-up were observed, while almost no Hunching was detected (Supplementary Fig. 2g–i). This result could be due to the different context of green light, which induces a stronger light response than red light does.

To determine whether optogenetic activation would be sufficient to inhibit Hunching, the same driver was used to express CsChrimson in these neurons, and red light was applied while simultaneously delivering an air puff. Optogenetic activation resulted in fewer Hunches and Static Bends (Supplementary Fig. 2j–l) upon air-puff responses. Moreover, if an air puff was delivered 3 s after light onset, no Hunching was observed at all (Fig. 1g–j, Supplementary Fig. 2m, n). Interestingly, R11A07>CsChrimson optogenetic activation triggered an escape response involving C-shapes and Rolls (Fig. 1g–j and Supplementary Fig. j–l), which is typically elicited in response to noxious stimuli[18,21–23]. These results suggested that the escape sequences triggered by R11A07 optogenetic activation could be due to the nociceptive inputs these neurons receive directly or indirectly.

To address this possibility, we analyzed all the inputs to A19c using EM connectomics[36] and found that both Basin-2 and Basin-4 receive nociceptive inputs from Multidendritic class IV neurons (Md IV), representing nearly a quarter of all the A19c inputs, whereas no significant input from Basin-1 was observed (Fig. 1c, d; Supplementary Fig. 1; Source Data 1). We previously showed that Basin-4, similar to Basin-2, inhibits Hunching and is required for Bending[17]. In addition,

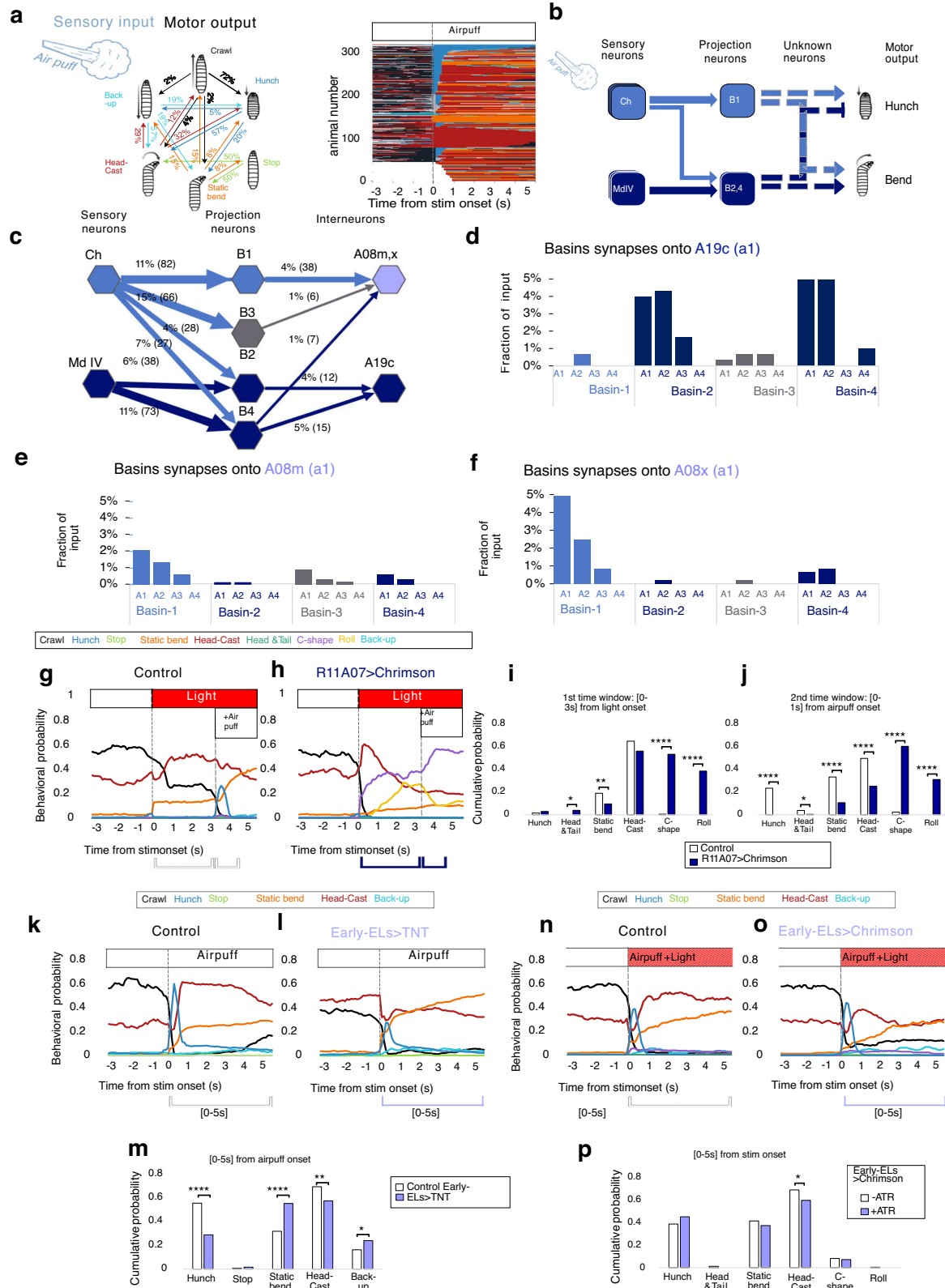

optogenetic activation of Basins triggers Rolling[18]. The above observation that silencing A19c resulted in a similar behavioral phenotype as silencing its main presynaptic partners, Basin-2 and Basin-4, raises the possibility that Basin-2 and Basin-4 inhibit Hunch and promote escape behaviors (i.e., Rolling) through A19c.

We further investigated the behavioral role of A08m,x by silencing early-born EL during air puff responses using TNT or GtACR1 and found that they are required for Hunching. (Fig. 1k–m, Supplementary Fig. 3a–e). Furthermore, optogenetic activation of early-ELs while simultaneously applying air puff resulted in a slight, although not significant, increase in Hunch probability in response to an air puff (Fig. 1n–p, Supplementary Fig. 3f, g). A08m/x could thus promote Hunching, as their presynaptic partner is Basin-1. It is however, important to note that the inactivation and optogenetic activation

**Fig. 1 | Second order interneurons in a mechanosensory network are differentially involved in responses to air-puff. a** Left: Transition probabilities in response to air puff (from Masson et al., 2020). Right: Ethogram. One line represents one individual, colors represent actions as in the schematic. To ease reading, larvae were grouped by their first behavioral response. attP2>TNT (*n* = 818). Some larvae were not tracked before the onset of stimulation (white space) (**b**) Simplified schematic of the previously characterized circuitry underlying Hunch/Bend (Jovanic et al., 2016). Ch: chordotonal neurons, Md IV: multidendritic class IV neurons, B1,B2,4: Basin-1, Basin-2 and Basin-4 (**c**). Synaptic connectivity based on EM reconstruction, between neurons in (**b**)., and A08m,x and A19c. Fraction of total input is shown. Light blue: all neurons previously identified as required for Hunch, dark blue: all neurons previously identified as inhibiting Hunch. B3 is in gray. Neurons in neuromere a1 are shown. **d** Input A19c receives from Basins across neuromeres a1-4, (**e, f**) Input A08m and A08x receive from Basins across neuromeres a1-4, Fractions of all input is shown (in a1). **g–j** Responses to light then air puff (3 s after light). **g** control (attP2>CsChrimson, *n* = 177) (**h**) larvae with activated R11A07 neurons (R11A07 > -CsChrimson, *n* = 250). **i** behavioral probability cumulated over the first three seconds after light onset: control (white) and R11A07 > -CsChrimson larvae (dark blue) (**j**). Same as (**i**) but over the first second after air puff onset, **k–m** Responses to 4 m/s air puff (**k**) control (eELGAL4>CantonS *n* = 254) (**l**). larvae with eELs interneurons inactivated (eEL-Gal4>TNT, *n* = 420). **m** behavioral probability cumulated over the first five seconds after air puff onset, control (white) and eEL-Gal4>TNT (lavender) (**n–p**) Responses to 4 m/s air puff and optogenetic activation (0.3 mW/cm² irradiance). **n** control (eEL-Gal4>CsChrimson, without ATR *n* = 248), (**o**). eEL-Gal4>CsChrimson, with ATR *n* = 286 (**p**) behavioral probability cumulated over first five seconds after air puff onset, eEL-Gal4>CsChrimson, without ATR (white) and with ATR (lavender). **g,h k, l, n,o** mean behavioral probability over time, Stim at 60 s. For all barplots: *: $p < 0.05$, **: $p < 0.005$, ***: $p < 0.0005$, ****: $p < 0.0001$, Chi² test, two sided. The source data and *p* values are provided in Source Data 1, 2 and 5.

experiments (Fig. 1k–p, Supplementary Fig. 3) involved all five early-born ELs (as targeted by the genetic line EL-Gal4,R11F02-Gal80), and of these five neurons, only A08m and A08x received significant input from the Basins.

Together, these results suggest that A19c and early-ELs A08m,x are second-order interneurons

that could be differentially involved in Hunching and escape behaviors. The early ELs A08m,x downstream of Basin-1 are required for Hunching, whereas A19c, which is downstream of Basin-2 and Basin-4, can inhibit Hunching and promote an escape sequence.

### Automated classification reveals context-dependent stereotyped escape sequences

We further characterized in detail the escape sequence triggered by optogenetic activation with the R11A07 driver. We found that, upon R11A07>CsChrimson optogenetic activation, the Roll was preceded by a C-shape, as has been previously reported in the nociceptive escape sequence[23]. We also observed that some larvae initiated the escape sequence with a symmetrical contraction that resembled a Hunch but consisted of both Head-and-Tail contractions before transitioning into a C-shape (Fig. 2). Indeed, the decrease in larval length was greater upon optogenetic activation with R11A07 than in Hunching during air puff stimulation (Supplementary Fig. 4a, b). This behavior has not been previously described as being part of the Rolling escape sequence.

To quantify the probability of different actions in the escape sequence under different stimulation conditions, we expanded machine learning-based behavioral classification algorithms[19] to detect Head-and-Tail and C-shape actions (Fig. 2a). The new classification was computed using an extra layer of random forest on top of the previous classifier, adding new sets of features and training on newly annotated data (see the detailed description in the Methods section).

Using these new classification algorithms, we examined the behavior sequence in response to an air puff, optogenetic activation of R11A07 neurons, or both (Fig. 2b–e, Supplementary Fig. 4). We found that larvae predominantly responded to an air puff with either a Hunch, a Static Bend, or a Head Cast[35,37] (Fig. 2d). Upon optogenetic activation of R11A07 alone, most larvae performed a C-shape that was often followed by a Roll (Fig. 2b, c, f, g; Source Data 3). In some larvae, the C-shape was preceded by a Head-and-Tail contraction. (Fig. 2b, c, f, g; Source Data 3). When optogenetic activation was combined with an air puff, the larvae performed Hunches and Static Bends in addition to the escape action sequences (Fig. 2d, e, h, i). Among the escape actions (Head-and-Tail, C-shape and Roll), relatively more Head-and-Tails and fewer C-shapes and Rolls were performed than after optogenetic activation alone (Fig. 2b–e, Source Data 3). These results reveal that different types of escape sequences are induced by R11A07>CsChrimson activation depending on whether it is coupled with mechanical stimulation or not: a Head-and-Tail > C-shape sequence is more prominent when the activation is coupled with an air puff, whereas the C-shape > Rolling sequence predominantly occurs after optogenetic activation alone (Fig. 2b, c). This raises the question of how the sensory context shapes the dynamics of nociceptive escape sequences.

### Sensory context modulates the competition between startle and escape actions and different types of actions within the escape sequence

To investigate the influence of mechanosensory stimulation on escape sequences triggered by optogenetic activation using the R11A07 driver, we subjected R11A07>CsChrimson larvae to different intensities of air puff and light stimulation (Fig. 3, Supplementary Fig. 5).

In the absence of an air puff, we observed distinct escape sequences depending on the level of optogenetic activation (Fig. 3). Head Cast and Crawls were prominent at low light intensity (0.1 mW/cm²), with very little C-shape or Rolling (Fig. 3a–c, j–m, v–x). While Head Cast and Crawl occurred in response to light, R11A07>CsChrimson larvae showed increased Head Casting and decreased Crawling compared with those of the control larvae after optogenetic activation (Supplementary Fig. 5a–c). The increase in Head Casting could be interpreted as a mild escape response triggered by weak optogenetic activation. A medium-high light intensity (0.2 mW/cm²) resulted in significantly more C-shapes and Rolls at the expense of Crawling (Fig. 3d–f, n–q, and v–x; Supplementary Fig. 5d–f). Increasing the light intensity further drastically increased the C-shape probability at the expense of Head Casting and, surprisingly, of Rolling (Fig. 3g–i, r–u, v, Supplementary Fig. 5d–i). In addition, at high light intensity, the Head-and-Tail probability increased significantly (Fig. 3g–i, r–u, v–x). These results suggest that different levels of optogenetic activation give rise to different escape sequences (Supplementary Fig. 5j). Weak optogenetic activation triggers mild escape responses composed mainly of Head Casts (sometimes followed by Crawls) and low levels of C-shapes and Rolls, whereas medium and high levels trigger an escape sequence. The escape sequence consists mostly of C-shapes, followed by Rolls for medium-intensity activation and Head-and-Tail and C-shapes for high-intensity activation. The Crawls observed after R11A07 optogenetic activation are faster than those observed in the control and therefore represent the Fast Crawl escape (Supplementary Fig. 4c).

To determine how mechanosensory information affects these escape sequences, we delivered air-puff stimulation of two different intensities, i.e., medium and strong, along with different levels of optogenetic activation. The air puff modulated the escape sequences triggered by optogenetic activation: Rolling was inhibited by a strong air puff at all light intensities (Fig. 3m, q, u). This decrease in Rolling probability was accompanied by an increase in Head-and-Tail contractions (Fig. 3m, q, u). A medium air puff was also sufficient to inhibit Rolling (and to promote Head-and-Tail contractions) at low and

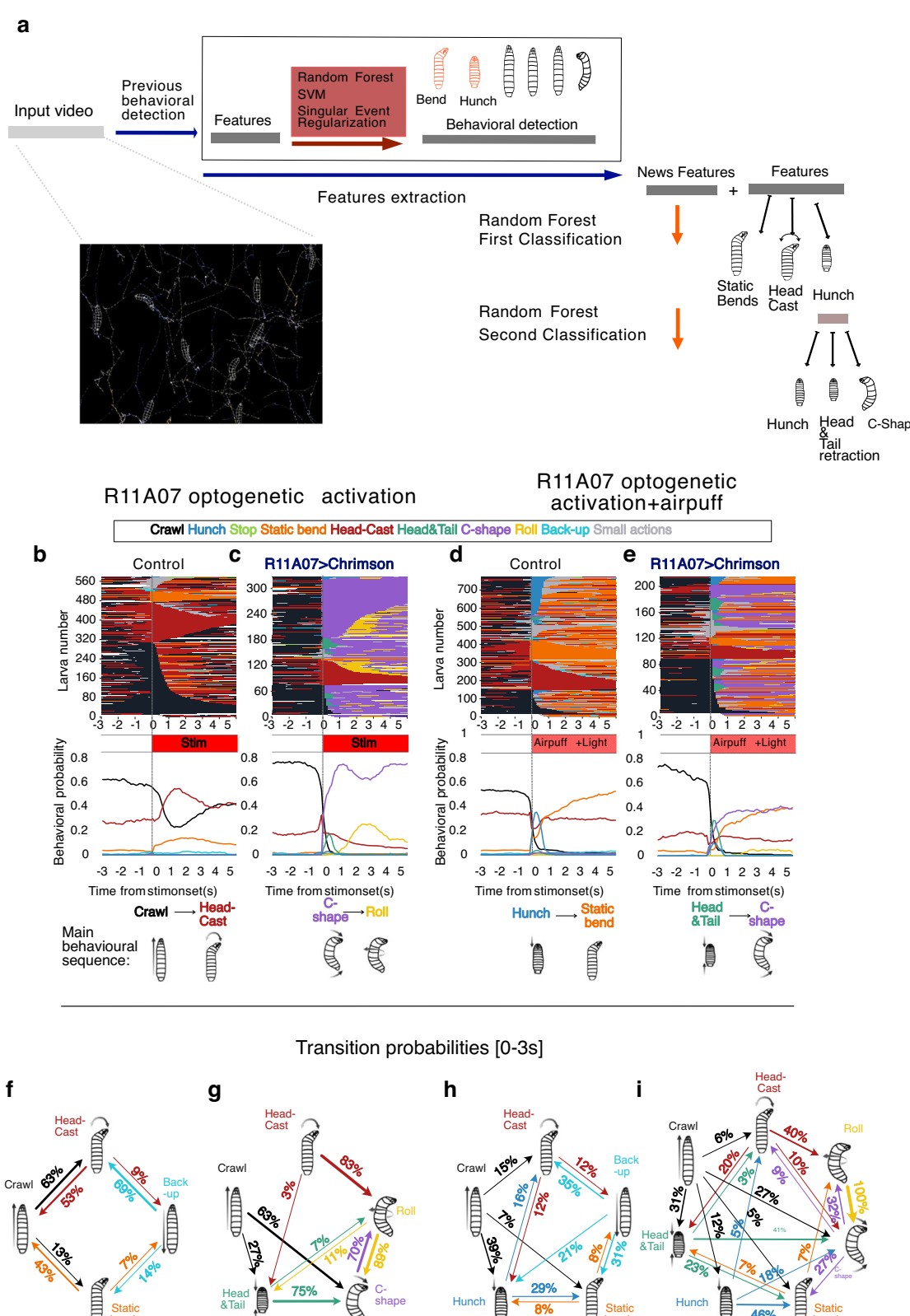

Transition probabilities [0-3s]

medium light intensities. This intensity-dependent increase in Head-and-Tail contractions and decrease in Rolling in the presence of air puff stimulation suggest context-dependent competitive interactions between different actions within the escape sequence.

Overall, adding air puff stimulation inhibited Roll, C-shape and Head Casts in favor of Head-and-Tails but also Hunches and Static

Bends (Fig. 3j–x). The different behavioral actions that larvae perform in response to somatosensory stimuli can be categorized into static, startle-like actions (Stopping, Hunching, and Static Bending) and active-type, exploratory and escape actions (Crawling, Head Cast, Fast Crawl, Head-and-Tail contractions, C-shape, and Roll)[17–19,22,23,26,38]. Thus, in addition to the competition between different actions of the escape

**Fig. 2 | An Automated classification of Bend-like behaviors reveals stereotyped escape sequences triggered by optogenetic activation of R11A07 neurons.**
**a** Reclassification procedure of larval behaviors automatically detected by a Machine Learning-based algorithm. Actions previously categorized as "Bend" were reclassified to be categorized as either "Head Cast","Static Bend" or "C-shape", and actions previously categorized as "Hunch"(head-retraction) were reclassified to be categorized as either "Hunch", "Head-and-Tail" or "C-shape". **b–e** Larval responses to R11A07 optogenetic activation (**b**) only light (control: attP2>CsChrimson, $n = 530$) (**c**). only light (optogenetic activation, R11A07>CsChrimson, $n = 305$). Upon optogenetic activation of R11A07 alone, most larvae perform a C-shape (63% out of larvae crawling). 27% of larvae that were crawling perform a Head-and-Tail contraction followed by a C-shape C-shapes are in 70% of cases followed by a Roll. d. air puff and light (control: attP2>CsChrimson ATR, $n = 765$) (**e**). air puff and light (optogenetic activation, R11A07> CsCrimsonn=214). Ethograms and mean behavioral probability over time are shown. Ethograms show actions over time, with one line corresponding to one individual and each color corresponding to a different action (as indicated). All larvae present between 59.8 and 61 seconds (for at least one time step) are displayed. For each condition a schematic of the characteristic behavioral sequence is depicted (**f–i**). transition probabilities cumulated over the first three seconds after stim onset (**f**). Control, light alone. **g** R11A07c>CsCrimson, light alone. (**h**) Control, air puff and light (**i**). R11A07>CsChrimson, air puff and light. When optogenetic activation is combined with air puff, the larvae performed relatively more Head-and-Tails (30%) and less C-shapes and Rolls, compared to when optogenetic activation was applied alone. Fewer larvae (45% compared to 67% upon optogenetic activation alone) perform C-shapes. In most cases (69%) Head-and-Tails were followed by a C-shape. C-shapes are less frequently followed by a Roll in presence of air puff (30%) than when the light was delivered alone (70%). Only transition probabilities of 3% or more are shown. Stim onset at 0 s. air puff intensity: 4 m/s. Irradiance: 0.3 mW/cm². Source data are provided in Source Data 3.

sequence, the air puff during optogenetic activation induces competition between startle-type and escape-type actions. The Static Bend and Hunch probabilities increase with increasing air puff intensity at low and medium optogenetic activation (Fig. 3a–f, j–q, v–x). At high levels of optogenetic activation, the Static Bend probability is increased at both air puff intensities, whereas Hunch is increased only at a medium air puff intensity (at the expense of Head-and-Tail contractions) (Fig. 3g–i, r–u, Supplementary Fig. 6). These data suggest that there could be competition between the mechanosensory-induced startle response and escape behaviors triggered by optogenetic activation of R11A07 neurons.

The outcome of the different types of competitive interactions (between the different actions in the sequence and between escape and startle behaviors) depends on the relative level of activation of R11A07 neurons and the air puff stimulus intensity. These findings suggest that these competitive interactions occur at the level of R11A07 neurons. In addition, the competition between actions induced by mechanosensory and nociceptive stimulation is nonhierarchical, and nociceptive and mechanosensory responses can mutually inhibit each other.

### R11A07 neurons are required for mechano-nociception

To determine whether A19c labeled by R11A07 is also required for nociceptive behaviors, we inactivated them using an inwardly rectifying potassium channel Kir2.1 while delivering nociceptive stimuli and monitored larval behaviors. Upon application of a mechano-nociceptive stimulus with a 50 mN calibrated *von Frey* filament, larvae performed either a C-shape bend, a Roll or a Turn (nonresponse) (Fig. 4a). Compared with the genetic controls, the larvae in which the R11A07-labeled neurons were silenced exhibited decreased nociceptive responses, specifically Rolling (Fig. 4a). In addition to nociceptive mechanical stimulation, Rolling has been shown to be triggered by local exposure to temperatures above 40 °C, and this thermo-nociceptive Rolling is also mediated by Md IV neurons[21,39]. We thus investigated larval thermo-nociceptive responses by applying a local heat stimulus using a temperature-controlled hot probe (46 °C) and measured larval Rolling response latencies. Silencing R11A07 neurons did not affect thermo-nociceptive responses (Fig. 4b). Thus, R11A07 neurons are required for mechano-nociceptive but not thermo-nociceptive behavioral responses.

The A19c neuron, labeled by R11A07, receives nociceptive input via Basin-2 and -4 neurons (Fig. 4c) and is thus likely to promote escape behaviors in response to nociceptive stimuli and inhibit startle behaviors in response to mechanosensory stimuli.

### Thoracic and abdominal neurons in the R11A07 line have distinct presynaptic connectivity

The R11A07 driver labels two types of neurons: abdominal A19c and a thoracic descending neuron (TDN) (Fig. 5a). To better understand the contributions of each neuron to the phenotypes described thus far, i.e., triggering escape sequences, inhibiting Hunching and participating in context-dependent competitive interactions, between startle actions and the escape sequence, we investigated the presynaptic connectivity of these two neurons.

The TDN sends descending lateral projections spanning nearly the entire length of the ventral nerve cord (VNC) in the R11A07 driver line (Fig. 5a–d). We identified a neuron with similar morphological features in the EM images. Compared with the abdominal A19c axon, the axon of this thoracic neuron is more lateral and dorsal (Fig. 5a–h). Both neurons, however, have axonal projections that connect to the contralateral side while receiving ipsilateral input with respect to the cell body (Fig. 5h–k). This suggests that they may play a role in controlling motor action requiring asymmetric contractions, such as Bends, C-shapes, and Rolls[24,40].

We further analyzed all the inputs of the TDN (Fig. 5l, m, Fig. 6a–d, Source Data 1) and found that the TDN receives inputs from the chordotonal subtypes lch5-2/4 in multiple neuromeres (a1, and a3-a8) (Fig. 6a, c, Source Data 1). These findings suggest that the TDN could integrate mechanosensory input across segments. By analyzing the distribution of all presynaptic partners of the neuron, we found that the mechanosensory input represents 15% of all inputs. Other significant fractions of inputs are from descending neurons (31%), from local neurons in the VNC (21%), and from ascending neurons (12%) (Fig. 5l, Fig. 6a, c). Most of the descending input the TDN receives from Brain and SEZ neurons consists of axo-axonic connections (Fig. 6a, c, Source Data 1) along the VNC, suggesting that higher-order neurons modulate the output of the TDN depending on context, state, or experience. In contrast, A19c receives most of its inputs from local and projection neurons in the VNC (47%), out of which Basins represent more than half (Fig. 5m). Only 8% of all inputs are from descending neurons, and A19c does not receive any direct inputs from sensory neurons (Fig. 5m, Fig. 6b, d, Source Data 1).

### Thoracic and abdominal neurons in the R11A07 line have distinct functions in defensive behaviors

Using calcium imaging in intact larvae, we functionally tested the connections between mechanosensory and the TDNs and Basin-2 and A19c neurons. First, we optogenetically activated Basin-2 using CsChrimson and imaged the calcium responses in R11A07 neurons with GCAMP6s. In line with the synaptic connectivity data from EM, optogenetic activation of Basin-2 induced calcium responses in A19c but not in the TDN (Fig. 6e–f). We then monitored the response of the two neurons to a mechanical stimulus. The TDN responded strongly to the stimulation, whereas the A19c neurons did not respond at all (Fig. 6g, h). These results are consistent with the mechanosensory-to-TDN connections and the absence of direct mechanosensory inputs to A19c that we observed in the EM reconstructions. However, A19c does receive inputs from Basin-2, which

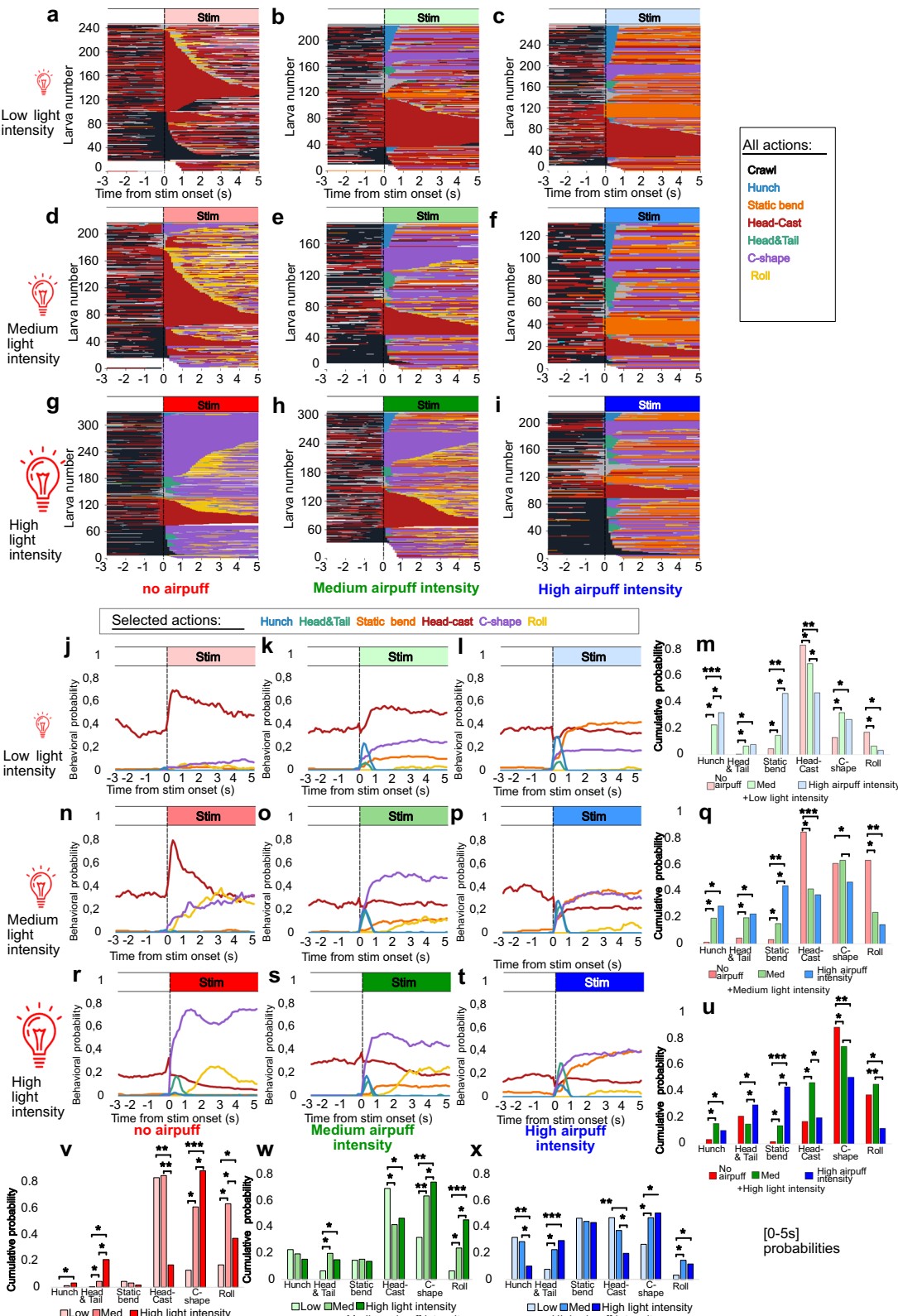

**Fig. 3 | Sensory context modulates escape responses triggered by R11A07>CsCrimson optogenetic activation. a–i** Ethograms including all detected actions. **j–x** behavioral probabilities for Hunch, Head-and-Tail, C-shape and Roll in response to: **j–m.** low light intensity (0.1 mW/cm²) with (**j**). no air puff (*n* = 199). **k** 3 m/s air puff (*n* = 206). **l** 4 m/s air puff (*n* = 269). **n–q** medium light intensity (0.2 mW/cm²) with (**n**). no air puff (*n* = 183) (**o**). 3 m/s air puff (*n* = 175). **p** 4 m/s air puff (*n* = 129). **r–u** high light intensity (0.3 mW/cm² irradiance) and with (**r**). no air puff (*n* = 305) (This data is the same as the one in Fig. 2c, but is used here

to compare optogenetic activation alone with combined optogenetic activation and air puff condition). r 3 m/s air puff (*n* = 265). **t** 4 m/s air puff (*n* = 214) Rolling is inhibited by both medium and strong air puff at low and medium light intensities, while at high light intensity is inhibited by only strong air puff. **m, q, u, v, w, x** Barplots correspond to the behavioral probability cumulated over the first five seconds after stim onset. For all barplots: FDR: *:*p* < 0.05, **:*p* < 0.01, ***:*p* < 0.005 Chi² test, two sided, with Benjamini-Hochberg correction. The source data and *p* values are provided in Source Data 5.

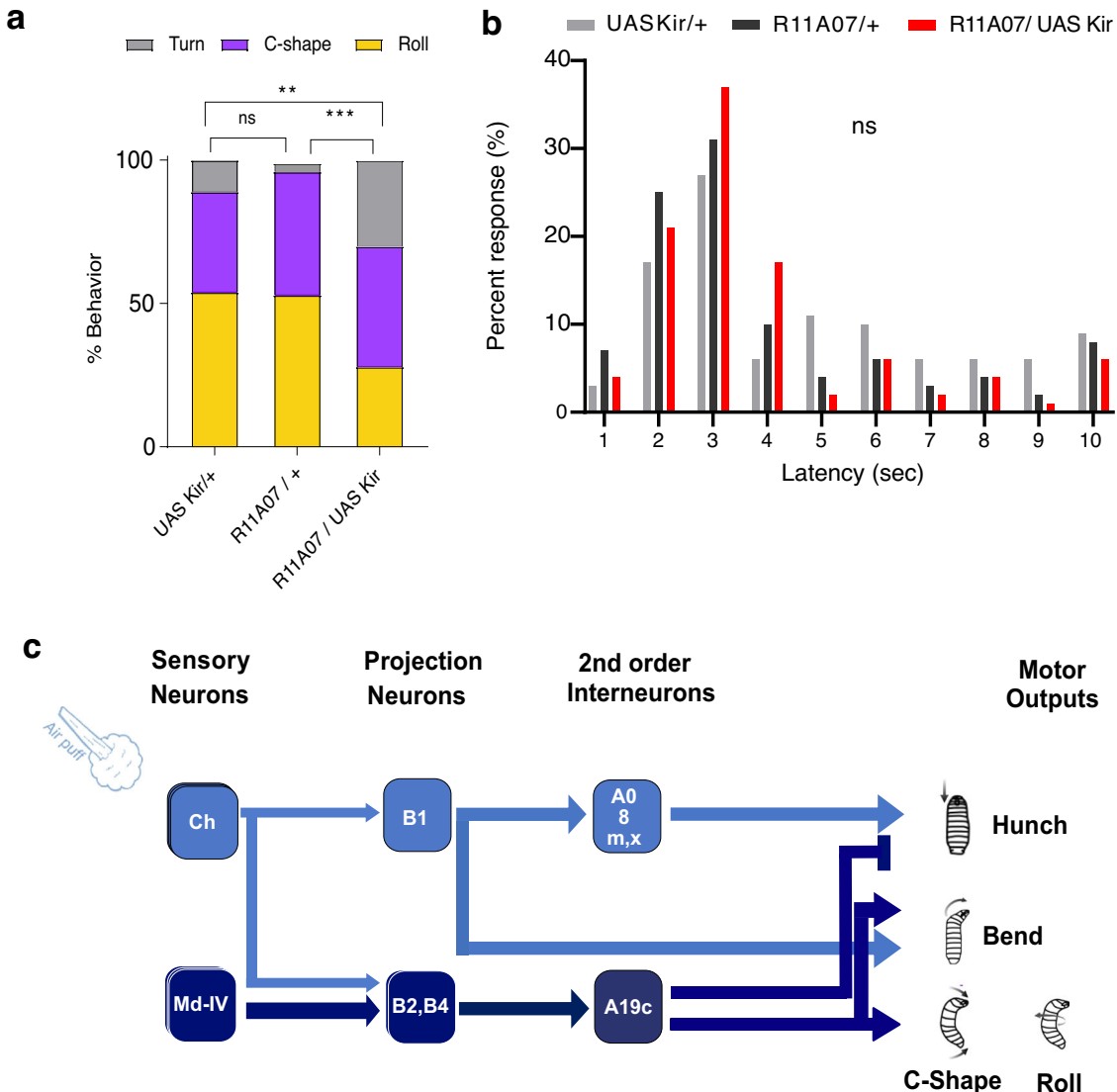

**Fig. 4 | The R11A07 neurons are required for mechano-nociception. a** Silencing R11A07 neurons using KIR results in less C-shape and rolling behavior in response to a mechano-nocipetive stimulus compared to the controls ($p = 0.0002$, $p = 0.0049$), $n = 63$ (UAS KIR/ +), 60 (R11A07/ +), 60 (R11A07/KIR) animals (**b**) Silencing R11A07 didn't affect latency of responses to a thermo-nociceptive stimulus, $n = 70$ (UAS KIR/ + ), 99 (R11A07/ + ), 90 (R11A07/KIR) animals. Chi-square test, two-sided, was used to compare behavioral probabilities in (**a**). \*\*:<0.01, \*\*\*: <0.001, Kruskal-Wallis test two-sided was used to compare latencies in (**b**). **c** Simplified diagram of the circuit model.

receive mechanosensory inputs[17,18]. The lack of responses in A19c leads to the hypothesis that A19c could be inhibited by neurons receiving mechanosensory inputs.

To determine which of the two types of neurons, A19c and TDN, promote escape actions, we used SPARC[41] to stochastically drive CsChrimson in subsets of the neurons in the R11A07 line (Fig. 7a–c). Among the 25 larvae tested that responded to optogenetic activation, four performed a Roll, four a C-shape, and five Hunched (Fig. 7d, Supplementary movie 1). The remaining larvae performed actions consistent with the response to the light stimulus. We dissected larvae that showed different phenotypes and used immunohistochemistry to reveal the neurons that were activated. In all the larvae, Hunched CsChrimson was expressed in at least one pair of TDNs and very few or no abdominal neurons. In the larvae that performed a Roll, no TDN was labeled, while either all or several pairs of A19c neurons were labeled. In the larvae with the C-shape phenotype, several pairs of A19c were present, and either no or single unpaired TDNs were observed (Fig. 7a–c, Source Data 4). These results suggest that the TDNs play a role in triggering Hunch and that A19c neurons trigger escape

behaviors (Fig. 7e). This effect could be achieved either through excitation or disinhibition, as the TDN and A19c are glutamatergic (Supplementary Fig. 7a) and therefore can be inhibitory[42]. TDNs are also cholinergic and thus also excitatory (Supplementary Fig. 7b). Both neuron types were negative for the GABA neurotransmitter (Supplementary Fig. 7c).

Taken together, these results show that the A19c, and not the TDN, promotes escape actions in a context-dependent manner. In addition, A19c inhibits Hunching, whereas TDN promotes Hunching.

**Relative level of A19c neuron activation determines defensive behavior**

Our behavioral data (Fig. 3) pointed to competition between the escape and startle responses evoked by optogenetic activation of the R11A07-labeled neurons and a coinciding air puff. The greater the degree of mechanical stimulation relative to the level of optogenetic activation, the stronger the inhibition of escape responses (Fig. 3). Since the TDN, but not A19c, was activated by mechanosensory stimulation and A19c activation triggered escape actions while TDN

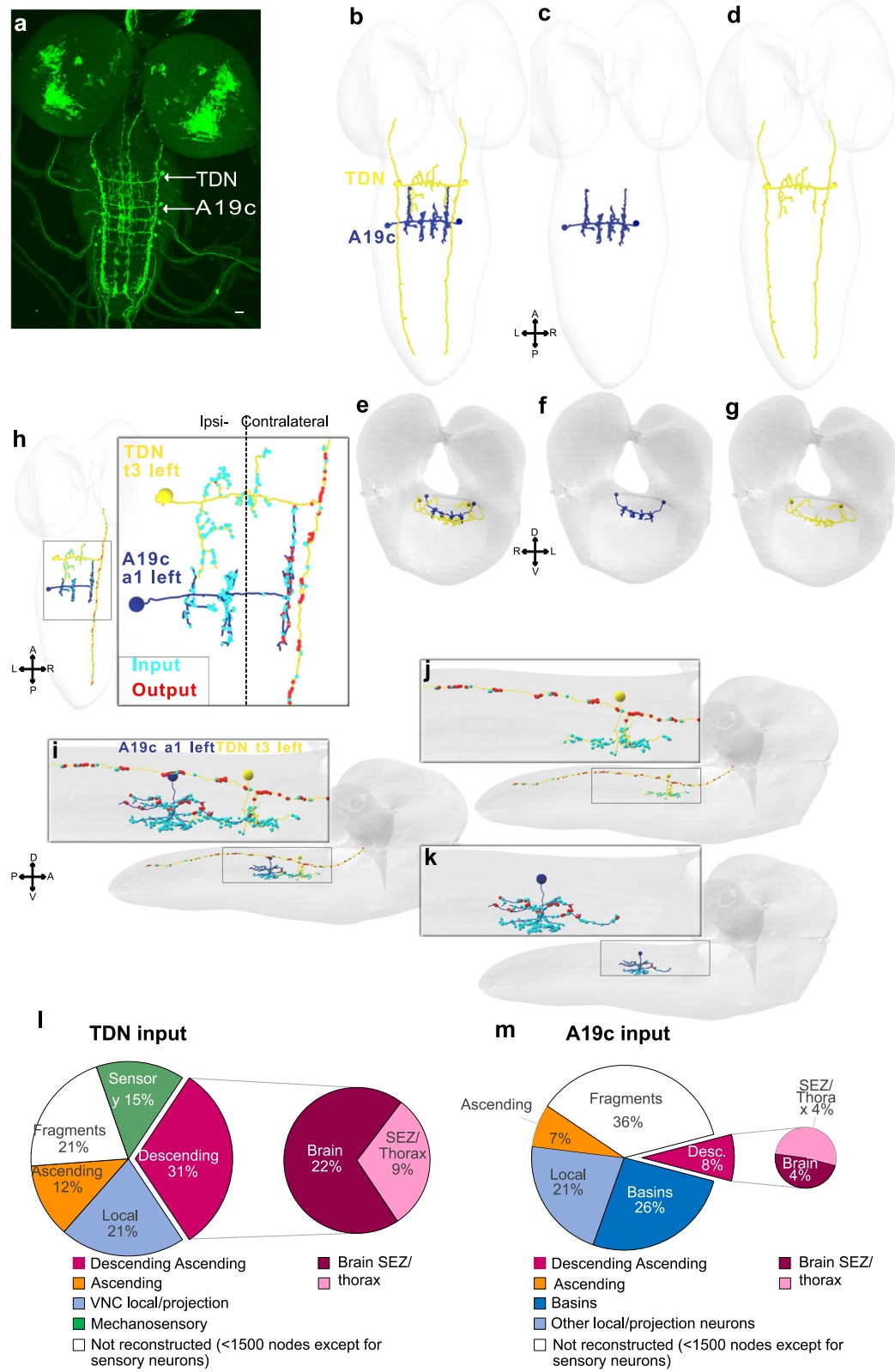

**Fig. 5 | Morphological characterisation of TDN and A19c neurons. a** R11A07 expression pattern (GFP). thoracic neuron in the segment T3, and four abdominal neurons in segments a1-4. Elsewhere in the CNS, expressing GFP in this line, are bundles of cell bodies likely corresponding to immature neurons. The expression patterns are consistent across all 11 larval brains that were imaged. Scale bar is 15 μm (**b,c**). Reconstructed A19c neuron in abdominal segment 1 (a1) (A19c, a1,left, right), in blue. **b,d** TDN, (t3, left,right) in yellow, antero-posterior view. **e–g** Reconstructed A19c (blue) and TDN (yellow) dorso-ventral view. **h** distribution of inputs (cyan) and outputs (red) in TDN_t3l and A19c_a1l, in an antero-posterior view. **i–k** same as in (**h**), viewed through a dorso-ventral angle. **j** shows the distribution of inputs for TDN (**k**) shows the distribution of inputs A19c (**l**). distribution of all input received by TDN (t3l,t3r) (**m**) distribution of all input received by A19c (a1l,a1r). In white are all inputs from neurons not reconstructed up to recognition, i.e. fragments. We considered all neuron skeletons with fewer than 1500 neurons to be unreconstructed, except for sensory neurons (in green). The source data are provided in Source Data 1.

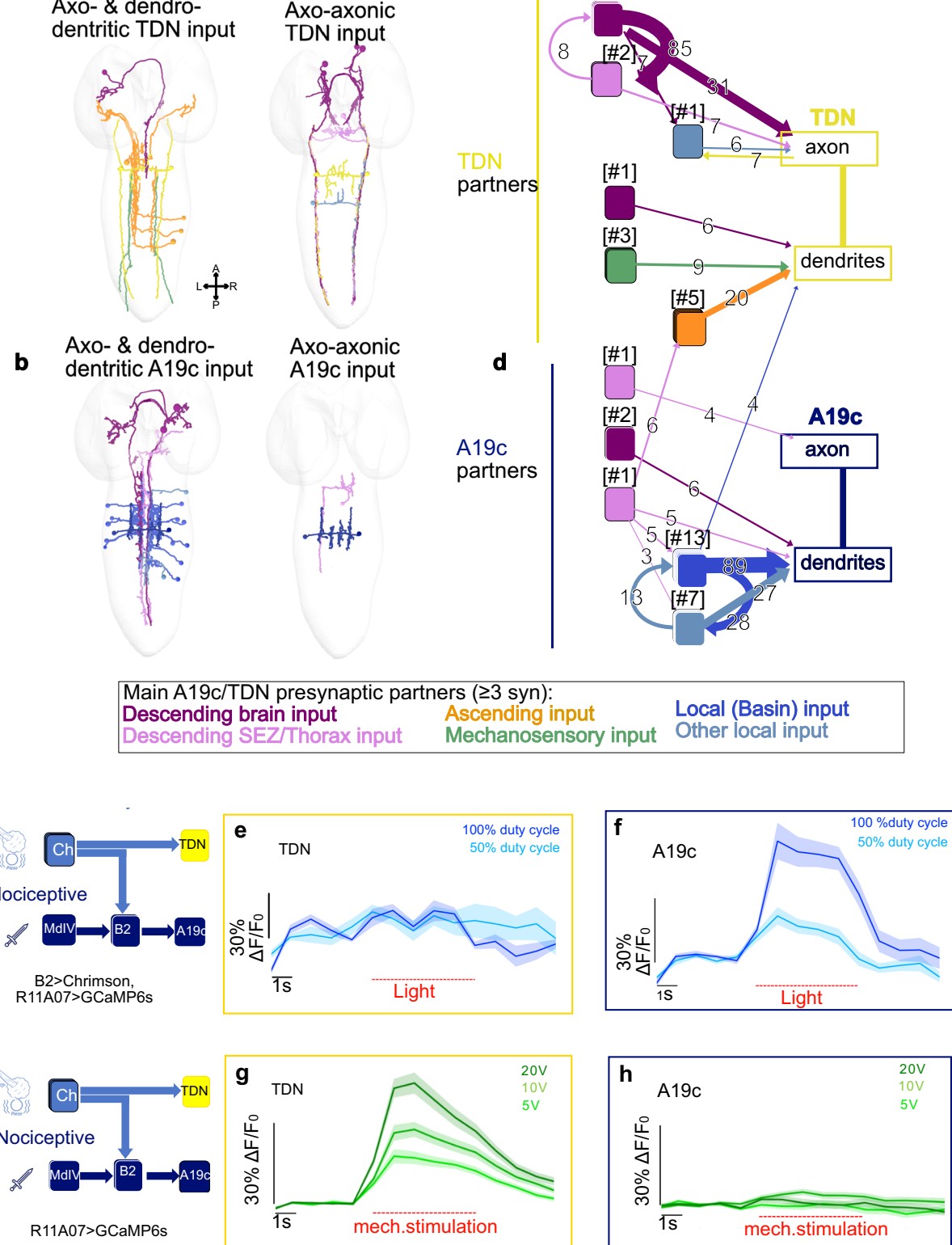

**Fig. 6 | TDN and A19c neurons receive distinct inputs. a–d** Main presynaptic partners (with 3 or more synapses) of TDN, A19c. Fragments were not included. **a** reconstructed skeletons in EM volume of TDN and its presynaptic partners (**b**). reconstructed skeletons in EM volume of A19c and its presynaptic partners. Presynaptic partners were coloured based on their localisation (brain, subesophageal zone (SEZ), sensory (cho), local, ascending neurons (**c, d**). Connectivity between neurons in (**a, b**). **a–d** axo- and dendro-dentritic synapses and axo-axonic synapses are shown All connections of 3 or more synapses are shown. Reconstructions were done in Catmaid software. **e, f** Live calcium imaging, using GCaMP6s, of TDN and A19c responses to optogenetic activation of Basin-2. **e** TDN response ($n = 2$ animals) (**f**). A19c response ($n = 3$ animals). **g, h** Live calcium imaging, using GCaMP6s, of A19c and TDN response to mechanical stimulation (piezoelectric-delivered vibrations) known to recruit chordotonal mechanosensory neurons (**g**). TDN response ($n = 10$ animals) (**h**). A19c response ($n = 8$ animals). Light and mechanical stimulations lasted 5 s. Mean and s.e.m are shown. The source data are provided in Source Data 1.

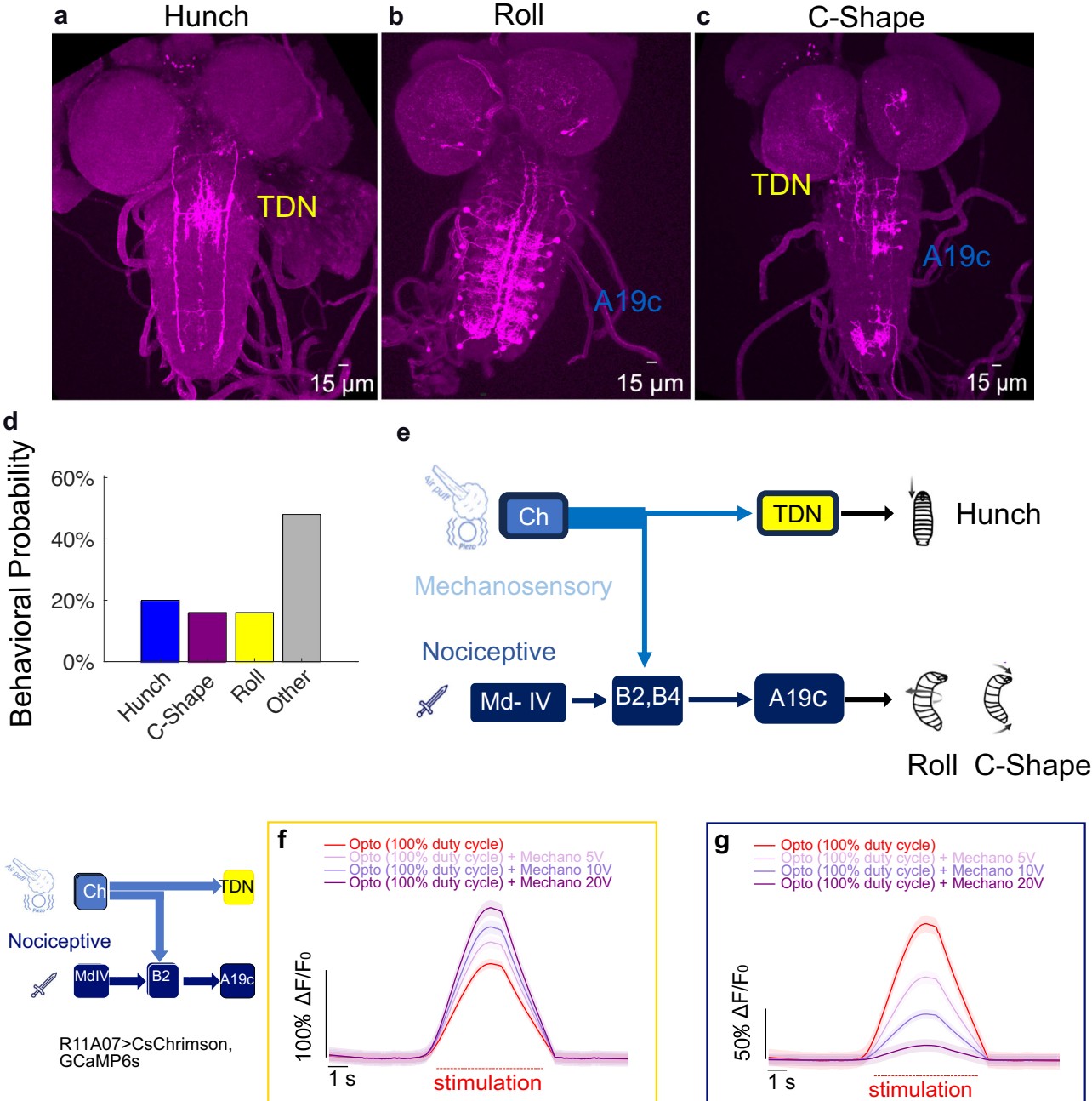

**Fig. 7 | A19c is sufficient to induce Rolling. a–c** Example of expression profile of CNS for larvae that (**a**). Hunched (n = 5), (**b**). Rolled (n = 3) or (**c**). Performed a C-shape (n = 5) (**d**). behavioral probabilities of larval responses to optogenetic activation. The source data are provided in Source Data 1 (**e**). Schematic showing that TDN promotes Hunching while A19c promotes escape behaviors. The genotype used were:(y,w;{nSyb-phiC31} attp5;R11A07-Gal4::UAS-SPARC2-I-CsChrimsontdtomato} CR-P40 and y,w;{nSyb-phiC31}attp5;R11A07-Gal4::UAS-SPARC2-S-CsChrimsontdtomato}CR-P40, n = 25 animals). **f, g** Live calcium imaging of A19c and TDN neurons upon their optogenetic activation and simultaneous presentation of mechanical stimulation (**f**). TDN responses (n = 8 animals) (**g**). A19c responses (n = 8 animals). Light and mechanical stimulations lasted 5 s. Mean and s.e.m are shown.

activation induced Hunches, we hypothesized that A19c neurons would be inhibited during mechanosensory responses. To test this hypothesis, we monitored the responses of TDNs and A19c neurons upon simultaneous optogenetic activation via the R11A07 driver and mechanical stimulation at different intensities (Fig. 7f–g, Supplementary Fig. 7d–g). As expected, the calcium responses in TDNs resulting from their optogenetic activation were facilitated by mechanosensory stimulation (Fig. 7f), whereas A19c activation was inhibited by mechanosensory stimulation (Fig. 7g). Similar to escape responses in behavioral experiments, this inhibition was dependent both on the intensity of stimulation and the level of activation of

A19c neurons. At constant levels of optogenetic activation, stronger mechanical stimulation leads to stronger inhibition. This inhibitory effect of mechanical stimulation on A19c neurons is weakened by increased activation of A19c neurons (Fig. 7g, Supplementary Fig. 7d–g). Thus, the sensory context modulates escape sequences by inhibiting A19c neurons in an intensity-dependent manner. A19c could thus be involved in competitive interactions with a neuron or neurons receiving mechanosensory inputs. The outcome of competition depends on the relative level of activation of A19c neurons by nociceptive pathways with respect to the level of activation of mechanosensory pathways.

## Selection of avoidance actions is gated downstream of Basin neurons in a context-dependent way

Projection neurons Basin-2 and Basin-4 (Fig. 1) inhibit the Hunch response to air puff[17] (Supplementary Fig. 8a–h). Since A19c received significant input from Basin-2 and Basin-4 (Fig. 1c–d), was activated by Basin-2 (Fig. 6e, f) and inhibited Hunch, similar to its presynaptic

partners (Fig. 1g–j, Supplementary Fig. 2), we hypothesized that Basin-2 and/or Basin-4 could inhibit Hunching through A19c. To test this hypothesis, we optogenetically activated Basin neurons while inactivating A19c with TNT using the R11A07 driver (Fig. 8, Supplementary Fig. 8i–l). As expected, in the context of intact A19c, optogenetic activation of Basin-2 or all Basins during larval responses to air puffs,

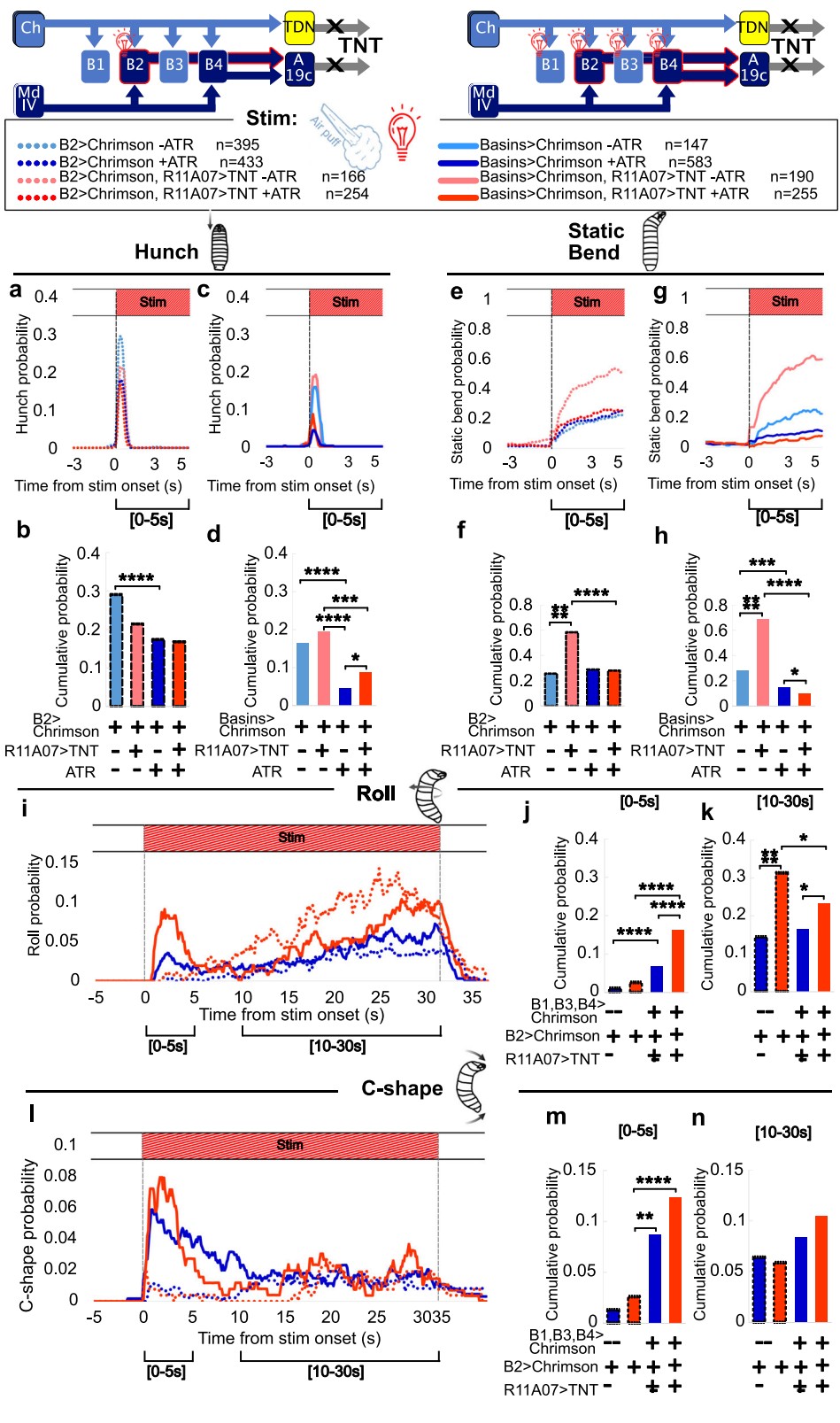

**Fig. 8 | Effect of Inactivating neurons in the R11A07 line on the behaviors triggered by optogenetic activation of Basins. a–d** Hunch responses. **a** Hunching probability over time, in response to air puff and optogenetic activation of Basin-2 (L38H09>CsChrimson, dark blue, dotted $n = 433$), and with R11A07 inactivated (L38H09>CsChrimson, R11A07 > TNT, red, dotted, $n = 254$). No All-Trans-Retinal (ATR) controls: L38H09>CsChrimson, light blue, dotted, $n = 395$ and L38H09>CsChrimson, R11A07 > TNT, light red, dotted, $n = 166$. **b** Hunching probability cumulated over the first 5 seconds after stim onset. Color code as in (**a**). **c** Hunching probability over time, in response to air puff and optogenetic activation of all Basins (L72F11>CsChrimson, dark blue, $n = 583$), with R11A07 inactivated (L72F11>CsChrimson, R11A07 > TNT, red, $n = 255$). No All-Trans-Retinal (ATR) controls: L72F11>CsChrimson, light blue, $n = 147$ and L72F11>CsChrimson, R11A07 > TNT, light red, $n = 190$. **d** Hunching probability cumulated over the first 5 seconds after stim onset. Color code as in (**c**). **e–h** Static Bend response. Color code and animal numbers as in (**a–d**). **e** Static Bend probability over time, in response to air

puff and optogenetic activation of Basin-2, with and without R11A07 inactivation. **f** Static Bend probability cumulated over the first 5 seconds after stim onset g. Static bend probability over time, in response to air puff and optogenetic activation of all Basins (L72F11) with or without inactivation of R11A07 neurons (**h**). Static Bend probability cumulated over the first 5 seconds after stim onset (**i–k**). Rolling probability in response to air puff and optogenetic activation of Basins, with and without inactivating R11A07, color code and animal numbers as in (**a–d**). **i** Rolling probability over time (**j**). Rolling probability cumulated over the first 5 seconds after stim onset. **k** Rolling probability cumulated over [10-30 s] within stim. **l–n** C-shape probability in response to air puff and optogenetic activation of Basins, with and without inactivating R11A07. Color code and animal numbers as in (**a–d**). **l** C-shape probability over time (**m**). C-shape probability cumulated over the first 5 seconds after stim. onset. **n** C-shape probability cumulated over [10–30 s] within stim. For all barplots: *: $p < 0.05$, **: $p < 0.005$, ***: $p < 0.0005$, ****: $p < 0.0001$, Chi² test, two-sided. The p-values are provided in Source data 5.

Hunch was inhibited (Fig. 8a–d, Supplementary Fig. 9, 10). The inhibition was stronger when all the Basins were activated than when Basin-2 alone was activated (Fig. 8a–d; Supplementary Fig. 9, 10). This may be because the combined activation of Basin-2 and Basin-4 activates A19c more strongly and thus has a greater inhibitory effect than the activation of Basin-2 neurons alone. The inactivation of R11A07-labeled neurons partially rescued the Hunching probability (but not the Static Bending probability) upon the activation of all the Basins but not upon the activation of Basin-2 alone (Fig. 8a–h, Supplementary Fig. 9h–k, Supplementary Fig. 10g–j, Source Data 5). This result suggests that the inhibition of Hunching (but not Static Bending) by Basins could be at least partly mediated by their postsynaptic partner A19c.

In addition to inhibiting Hunching, the optogenetic activation of Basins triggers Rolling[18]. Thus, we investigated whether A19c also mediates the escape behaviors triggered by Basin activation. Surprisingly, we found that expressing TNT in the R11A07 neurons increased Rolling and C-shapes upon Basins activation, both when air puff combined with light (optogenetic activation) (Fig. 8i–n) and when light alone was used (Supplementary Fig. 8i–l). Thus, the neurons in R11A07 not only trigger C-shapes and Rolling, as shown earlier (Figs. 1–3, Fig. 7) but also inhibit these behaviors upon Basin activation. Given that selective optogenetic activation of A19c neurons triggers escape behaviors (Fig. 7), this inhibition of Rolls and C-shapes is likely mediated by TDNs.

The optogenetic activation of all Basins and Basin-2 results mostly in C-shapes and an increase in Head Casting (Supplementary Fig. 8i–l, Supplementary Fig. 9a–g, Supplementary Fig. 10a–f). An examination of the head and tail speed of Head Cast events revealed 'Tail Casts', especially in larvae with optogenetically activated Basin-2 neurons (Supplementary Fig. 11a, b) and faster Head Cast (Supplementary Fig. 11 c, d). Examination of the Crawling speed prior to and upon optogenetic activation revealed Fast Crawls triggered by Basin activation (Supplementary Fig. 12). The speed increased upon the expression of TNT in R11A07 neurons. As discussed previously, Fast Crawls were also induced by optogenetic activation of R11A07 neurons (Supplementary Fig. 4). Thus, R11A07 neurons can inhibit both Rolling and Fast Crawling upon Basin optogenetic activation.

Overall, we found that neurons labeled by the R11A07 line can mediate the inhibition of Hunching by Basins (likely A19c), as well as inhibit C-shapes, Rolls and Fast Crawls triggered by Basin optogenetic activation (likely via the TDN).

## TDNs and A19c neurons connect to the premotor and motor layers through distinct pathways

Calcium imaging and SPARC experiments revealed that the TDN and A19c are differentially involved in startle and escape actions (A19c promotes C-shapes and Rolls, whereas the TDN promotes Hunching)

(Figs. 6 and 7). In addition, both the optogenetic activation and silencing of R11A07 neurons upon Basin activation induced Rolling. These findings suggest opposing roles of these neurons in Rolls. To investigate this effect, we analyzed the postsynaptic connectivity of the TDN and A19c (Fig. 9, Supplementary Fig. 13, Source Data 6) in the EM images. This analysis revealed that the TDN is directly connected to motor neurons, whereas A19c is not (Fig. 9a, Source Data 6). Since the TDN receives direct sensory input from multiple segments (Fig. 6, Source Data 6), this result suggests that the TDN can trigger behaviors in response to direct mechanosensory inputs. The main direct partners downstream of the TDN are the premotor neurons T19v and A19d. The premotor neuron T19v receives inputs from multiple neurons previously described as triggering Rolling, namely, Wave[43] and Basin-2, Basin-4 and pre-Goro neurons[18]. Moreover, T19v is the main premotor target of Wave and Basin-2 and Basin-4 (Supplementary Fig. 13, Supplementary Fig. 14). Thus, the TDN may inhibit Rolling by influencing T19v.

One of the main direct partners downstream of A19c synapses are the premotor neurons Marathon and A08e1 (Fig. 9a, Supplementary Fig. 13), the latter belonging to a group of late-born EL interneurons previously described to be involved in left–right symmetric muscle contraction[24].

The main direct targets of the TDN were either premotor (three neurons, 29 synapses) or motor neurons (two neurons, 16 synapses), with just one main target being an interneuron not connected to motor neurons, receiving 18 synapses from the TDN (Fig. 9a, Supplementary Fig. 13, Source Data 6). In contrast, A19c connects to more interneurons that are not directly connected to motor neurons (Fig. 9a, Supplementary Fig. 13, Source Data 6). These findings suggest that the TDN is more strongly connected to motor neurons than A19c is, both directly and indirectly.

The motor neurons targeted by the TDN (and its partners A19d and T19v) innervate broad muscle groups both dorsally and ventrally (the RP2 and RP5 muscles, respectively), as do the ventro lateral (VL), ventral oblique (VO) (i.e., VO 15, 16) and dorso lateral (DL) (9, 10) muscles. The A19c premotor neuron targets connect to MNs innervating primarily the DL (1, 9, 10) and DO (5,11,19) muscles. These DOs are not targeted by TDN partners. Two recent studies revealed muscle pattern activity during escape Rolling[40,44]. Silencing MN neurons innervating the VL and DO by Cooney et al. blocked Bending and Rolling, which is consistent with the role of the VL muscles in Rolling. In addition, Cooney et al. reported that silencing MNs innervating the DL reduced Rolling. He et al. identified DL9 as one of the 11 muscles that is robustly active during Rolling, whereas DL1 and DL10 are less robustly activated. He et al. described a total of 11 muscles belonging to the VL, VO, DLA and ventral acute (VA) groups that were active during Rolling. Among these 11 muscles, 6 (DL 9, 4, VL 12, VO 15, 16 and 28) were targeted by MNs receiving indirect

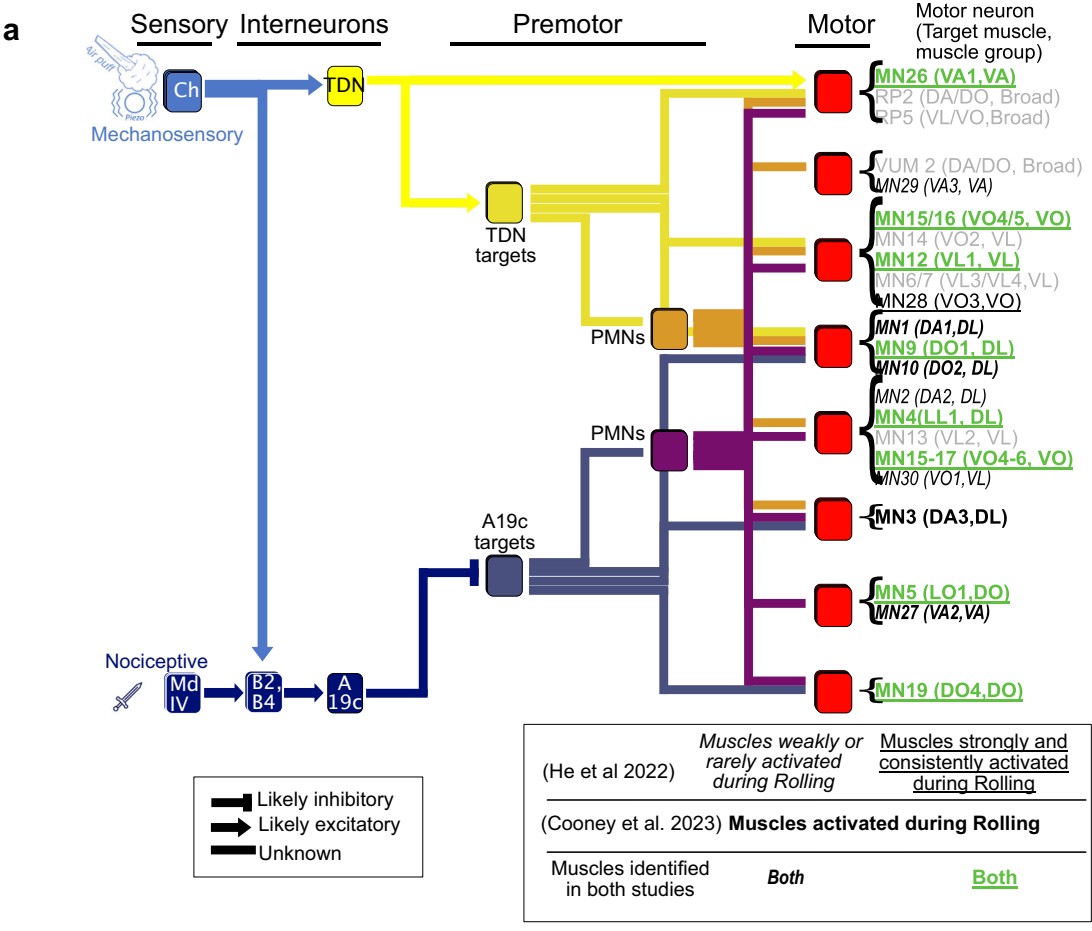

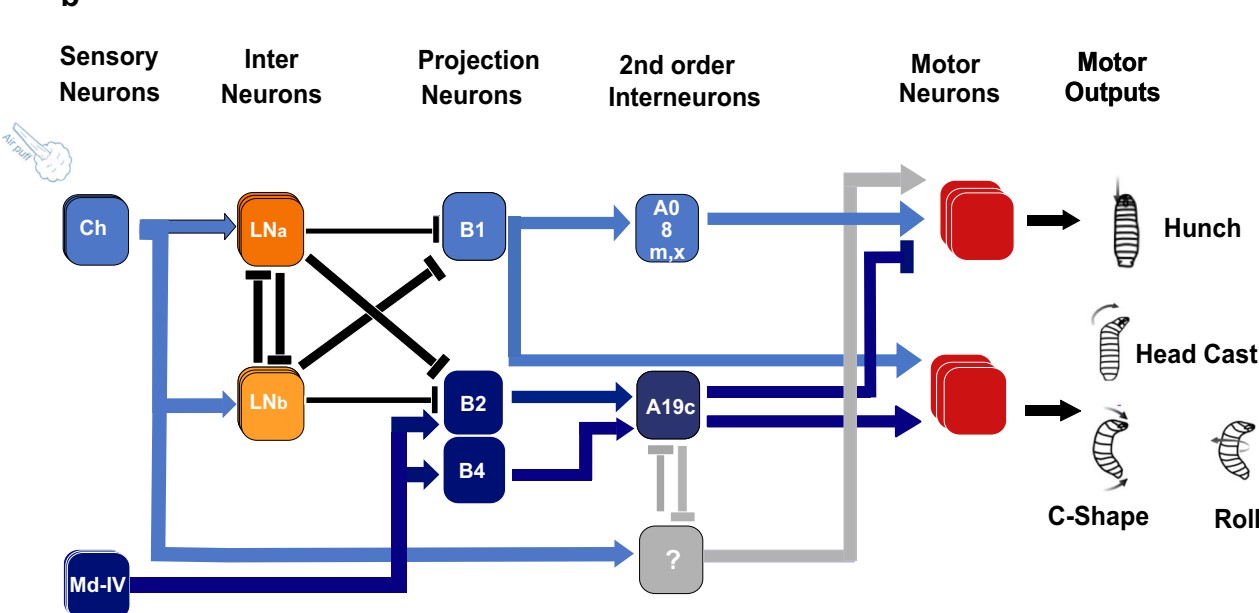

**Fig. 9 | TDN and A19c neurons connect to the motor side through distinct pathways. a** Schematic summarizing TDN and A19c connectivity, and their respective synaptic distance to specific motor neurons. This schematic is a simplified representation of the connectivity shown in Supplementary Fig. 13. Postsynaptic partners of TDN were grouped based on whether they were direct (yellow) or indirect (orange) targets of TDN. Similarly, A19c direct (blue) and indirect (purple) targets were regrouped. TDN and A19c contact, directly and indirectly, 22 motor neurons (MNs) located in segment A1. Note that, just as in Supplementary Fig. 13, the motor neurons from other segments were not taken into account. The 22 A1 MNs contacted by TDN and/or A19c were placed into 8 groups (red), depending on their synaptic distance to TDN and A19c. Importantly, among these 22 MNs are MNs targeting muscles identified as recruited during Rolling events, in two recent studies[40,44]. These "Rolling MNs" are coloured and highlighted accordingly. In gray are MNs controlling muscles found not to be activated during Rolling by either study, as well as MNs innervating broad sets of muscles (RP2, RP5 and VUM2 MNs). **b** Simplified diagram of the circuit model. The source data are provided in Source Data 6.

inputs from both A19c and the TDN, whereas 3 (doral oblique (DO) 5, 11, 19) were targeted only by neurons receiving indirect input from A19c.

Taken together, these connectivity analyses reveal distinct pathways for the TDN and A19c to motor neurons, with the TDN being closer to the motor side than A19c and the TDN recruiting motor neurons more broadly than A19c. Interestingly, both A19c and the TDN indirectly connect to the DL and VO muscles, which are putatively involved in Rolling, although through distinct premotor neurons. In addition, A19c indirectly connects to the DO-innervating MN involved in Rolling.

## Discussion

Using machine learning-based algorithms for behavioral classification combined with targeted neuronal manipulations, EM connectomics, and calcium imaging, we characterized the competition between defensive Startle (static) and Escape (active) defensive behaviors at the behavioral and neural circuit levels. We identified second-order interneurons that are differentially involved in startle and escape behaviors. Furthermore, we determined that the sensory context modulates defensive behaviors by influencing the relative level of activation of second-order interneurons that promote escape behaviors. Finally, from connectomics analysis, we identified a descending neuron that could be involved in modulating escape sequences depending on mechanosensory information and inputs from the brain.

To evade potential dangers, animals use diverse defensive strategies ranging from static protective actions to vigorous and fast active forms of escape behavior[5]. The selection of behavior that is performed depends on the type of danger as well as the context or animal's behavioral or internal state. The neural circuitry that ensures selection between these different options has not yet been extensively characterized, but competition between freezing and flight has been proposed to involve reciprocal inhibitory connections between two populations of inhibitory neurons[13]. Our previous work identified a similar motif in *Drosophila* larvae with synaptic and cellular resolution and mapped the circuit that underlies the competition between the two most prominent actions that occur in response to the air puff: Hunch, a passive type of response, and Bend, an active type of response[17]. In this circuit, Hunch is encoded by the activity of the projection neuron Basin-1 and the absence of Basin-2 activity at the output of the circuit. The coactivation of both neurons triggers Bend (and inhibits Hunch). Here, we identified the next layer in the sensory processing stage, which is composed of two groups of neurons. Early-born even-skipped lateral (EL) neurons that receive primarily Basin-1 (and Basin-3) inputs are required for Hunch, whereas A19c neurons that receive primarily Basin-2 (and Basin-4) inputs inhibit Hunch. Connectivity analysis did not reveal mutual direct or indirect connections between these neurons. Additional sites of competition could thus exist at later stages of sensorimotor processing.

EL interneurons have previously been shown to be involved in maintaining left–right symmetric contraction[24]. The early-born ELs that receive inputs from chordotonal neurons have been shown to respond to vibration[25] and have been proposed to be part of a mechano-nociceptive escape circuit as their optogenetic activation triggered Rolling escape[25], which we also observed at high optogenetic intensity (2.2 mW/cm² irradiance) and can be explained either by the input A08m,x received from Basin-2 and Basin -4, or by the implication of the other three early-born EL neurons in escape behaviors.

Our results show that early-born ELs are both required for and can facilitate air puff-induced Hunching. Thus, early-born ELs could be involved in different types of defensive behaviors induced by mechanical stimuli. We manipulated the early-born ELs using a driver that labels the five different early-born neurons. Two of these receive mechanosensory inputs (from chordotonals and Basin-1 and Basin-3 neurons), and three receive unknown inputs. Thus, similar to Basins, the different types of avoidance actions (active and passive forms of escape) may be encoded differently by different types or combinations of different types of early-born EL interneurons.

In this study, we also show that A19c inhibits Hunching and triggers an escape sequence that, in addition to the previously characterized C-shapes and Rolls[23], can also comprise Head-and-Tails as the first action in a sequence. It remains unclear whether Head-and-Tail is indeed a part of an escape sequence in some natural contexts or whether it occurs only after R11A07 optogenetic activation when A19c and the TDN are coactivated.

We also show that the relative levels of optogenetic activation of R11A07-labeled neurons and the air puff intensities determine which type of escape sequence is triggered. The R11A07-labeled A19c neurons are located downstream of the early processing stage competition site. This finding suggests that a site of competition between static and dynamic escape actions also exists at the level of second-order neurons. Since the connectivity analysis did not reveal any reciprocal connections between early-born EL neurons and A19c neurons or TDNs and A19c neurons, other neurons receiving mechanosensory inputs could be involved (Fig. 9b). Additional sites of competition could also exist at the premotor site, where competing neurons could be activated by mechanosensory stimulation and nociceptive stimulation.

Previous studies have shown that Basin-2 and Basin-4 neurons inhibit Hunching during air puff responses[17] and that the activation of Basins triggers escape behaviors[18]. In this study, activating Basin neurons while R11A07-labeled neurons were silenced resulted in a moderate increase in Hunching, suggesting that A19c could be involved in the inhibition of Hunching by Basin-2 and Basin-4 neurons. Furthermore, stochastically activated subsets of neurons in the R11A07 line confirmed that A19c neurons promoted Rolling, whereas TDNs promoted Hunching. Rolling is thought to be the fastest type of escape action induced by the most noxious stimuli[18,21,22]. Although increasing the level of optogenetic activation seemed to induce progressively stronger escape behaviors, from Head Casting at lowest intensities to Rolling at medium and high light intensities, the strongest optogenetic activation of R11A07-labeled neurons in most instances triggered C-shapes and not Rolling. This may be due to the uncoordinated effect of optogenetic activation (simultaneous activation of both the left and right sides in multiple segments) compared with natural stimuli, which may be unilateral and more local. In addition, increasing optogenetic activation could have a differential effect on TDN and A19c neuron activity or possibly on premotor neurons activated by the TDN and A19c and could thus trigger different behaviors. Additionally, the TDN is activated by mechanical stimulation, whereas A19c is not. Thus, combining mechanical stimulation with optogenetic activation of R11A07 could result in a relatively greater level of TDN activation than A19c. This greater level of TDN activation could result in sequences with more Head-and-Tails and fewer Rolls that were observed when optogenetic activation was combined with air puff. High levels of optogenetic activation also resulted in less Rolling and fewer Fast Crawls (Fig. 3). These findings suggest that A19c neurons can gate escape responses on the basis of context, depending on their level of activation. This dose-dependent gating of the different actions within the escape sequence is reminiscent of the DnB (Down and Back) neuron activation level-dependent C-shape and Rolling shown by Burgos et al.[23]. We propose that the sensory context could influence the level of activation of R11A07 neurons (by activating/inhibiting them), as suggested by the results of calcium imaging experiments. The R11A07 neurons would then gate the behavioral responses depending on their level of activation. Indeed, our results show that the Basin-2, Basin-4/A19c pathway is involved in Head Casting during the air puff response and in C-shape and Rolling during the mechano-nociceptive response. Thus, the same pathway promotes escape behaviors (at the expense of protective actions), in both a

mechanosensory and nociceptive context. The type of escape behavior expressed in different contexts could depend on the level of A19c activation.

Context is usually thought of as a combination of internal and external information that modifies the stimulus-response relationship[8,20,45]. In this study, the optogenetic activation of R11A07-labeled neurons triggered the escape sequence, and the presence or absence of air puff represents different contexts. However, in nature, the distinction between what is a stimulus and what is context is less obvious, and the additional information may be part of the stimulus itself that would induce a change in the behavioral response[7,8,45]. Indeed, it has previously been shown that the multisensory integration of mechanical and nociceptive stimuli enhances action-selection[18]. However, regardless of the definition, the investigation of the influence of "context" on stimulus-response pairing sheds light on how the different types and different combinations of information are processed and represented at the neural circuit level. Here, we found that the relative levels of activation of the A19c neurons and air puff intensities determined the probabilities of the different actions in the action sequence. Stronger air puff intensities inhibited Roll, whereas high optogenetic activation inhibited Hunch. These findings suggest that TDNs and A19c neurons could gate behavioral selection depending on contextual information toward more active or passive forms of avoidance behavior. This competition is mutual: the nociceptive pathway elicits active and fast forms of escape and inhibits protective Hunching and Static Bending; conversely, the mechanosensory pathway inhibits escape behaviors in favor of protective/startle-like actions. A recent study revealed that chordotonal activation can gate weak nociceptive inputs by acting on second-order A08n interneurons through GABAergic inhibition of nociceptive synaptic outputs[46]. Our results indicate additional sites of cross-modal gating of the nociceptive response by mechanosensory stimulation closer to the motor side. Additionally, our results do not point to unilateral gating but rather mutual competition between mechanical and nociceptive responses. This could be important if the animal is confronted with two types of dangers and needs to decide which aversive stimulus to attend to as a priority, depending on the degree of each danger.

In addition to the inputs from mechanosensory neurons, TDNs receive descending brain inputs through axo-axonic connections, suggesting that TDNs can modulate motor output on the basis of integrated multisensory information, state, motivational drive and/or experience.

The activation of R11A07-labeled neurons triggers a Roll, whereas the silencing of these neurons upon Basin activation results in more Rolling and Fast Crawling. Thus, A19c, TDN or both types of neurons could inhibit Roll and the Fast Crawl. Because the TDN does not receive input from Basin-2 and generally receives very weak input from other Basins, one possibility is that the inhibition of Rolling (and Fast Crawling) upon Basin activation could be mediated by the A19c neuron. However, the TDN could also inhibit Roll depending on inputs from the brain by acting on downstream neurons directly on the motor side (e.g., T19v neurons). Indeed, the TDN responds to mechanical stimulation and promotes Hunching (Figs. 6 and 7), and the inhibition of Roll and Fast Crawl is likely mediated by the TDNs.

A19c and the TDN connect indirectly to motor neurons innervating muscles shown to be involved in Rolling[40,44]. A19c connects (indirectly) to MNs that innervate "Rolling muscles", which are not targeted by TDN partners, suggesting that it could be more significantly involved in Rolling. Cooney et al. reported that muscle activity during Rolling is synchronous across segments and left-right asymmetric except for short periods of left-right hemisegmental symmetry when homologous muscles along the dorsal and ventral midline enter the bend and co-contract. They proposed that sequential firing of excitatory and then inhibitory premotor neurons (PMNs) on one side followed by the activation of their counterparts on the other

side could underlie the progression of muscle contraction waves around the circumference of the larva. Further functional investigations of the downstream partners of the TDN and A19c identified by the connectome will reveal how each of these neurons influences the patterns of motor activation underlying the Rolling behavior.

In conclusion, our findings reveal context-dependent competitive interactions between startle and escape actions implemented at the level of second-order interneurons. This site of competition that integrates external sensory information exists in addition to a previously identified site at the early processing stages for selection between a startle-type and an escape-type action that integrates internal-state information[17,47] (Fig. 9b). This finding suggests that the competitive interactions underlying the selection of appropriate defensive behaviors in response to threats could be distributed in the nervous system and may integrate various information about the internal state and the environment.

## Methods
### Animal rearing and handling
Flies (*Drosophila melanogaster*) were raised on standard food media (2% ethanol, 0.4% methylhydroxybenzoate, 8% yeast, 8% cornmeal, and 1% agar) at 18 °C. Third instar larvae were collected as follows: male and female flies from the appropriate genotypes were placed together for mating and then transferred to a Petri dish containing fresh food medium for egg laying at 25 °C for 12–16 h. The Petri dish was then placed at 25 °C for 72 h. Foraging third instar larvae were collected from the food medium using a denser solution of 20% sucrose, scooped with a paint brush into a sieve and gently and quickly washed with water. Larvae used for optogenetic experiments were raised at 25 °C in complete darkness on standard food supplemented with all-trans retinal (reference R240000, Toronto Research Chemicals) at 0.250 mM for the Chrimson experiments and 0.5 mM for the GtACR1 and SPARC-Chrimson experiments. The full list of genotypes used can be found in the Supplementary Method 1 Resource Table.

### Behavior tracking
To track behavior, we used a previously described apparatus[17,19,26]. Briefly, the apparatus comprised a video camera (Basler acA2040-90umNIR camera) for monitoring larvae, a ring light illuminator (Cree C503B-RCS-CW0Z0AA1 at 624 nm in the red), a computer and a hardware module for controlling air puff. The arena consisted of 25,625 cm$^2$ of 3% Bacto agar gel (CONDALAB 1804-5) with charcoal (Herboristerie Moderne, 66000 Perpignan) in a plastic dish and was changed for each experiment. For optogenetic experiments, plates without charcoal were used (in order to be transparent), and larvae were tracked using IR light from below (through the agar). The collected and washed third instar larvae were moderately dried and gently spread on the agar starting from the center of the arena using a soft-haired brush. Approximately 30–100 larvae were tested simultaneously during each experiment. The temperature of the behavior room was maintained at 25 °C. The larvae were tracked using the Multi-Worm Tracker (MWT) software (http://sourceforge.net/projects/mwt)[26,48].

### Air puff stimulation
The air puff was delivered as described previously[17,19,26] to the 25,625 cm$^2$ arena at a pressure of 1.1 MPa through a 3D-printed flare nozzle placed above the arena, with a 16 cm × 0.17 cm opening. The nozzle was connected through a tubing system to the plant to supply compressed air. The strength of the airflow was controlled through a regulator downstream from the air amplifier and turned on and off with a solenoid valve (Parker Skinner 71215SN2GN00) with consistent coverage of the arena across the experimental days. The air–current relay was triggered through TTL pulses delivered by a measurement computing PCI-CTR05 5-channel counter/timer board at the direction

of the MWT. The onset and duration of the stimulus were also controlled through the MWT. Larvae were allowed to crawl freely on the agar plate for 60 s prior to stimulus delivery. Air-puff was delivered at the 60th second and applied for 30 seconds. Larvae were also recorded 60 s after the end of stimulation. The air flow rates at 12 or 9 different positions in the arena were measured with a hot-wire anemometer (PCE-423, pCE instruments) to ensure consistent rates across experimental days. The air puff was triggered through TTL pulses delivered by a Measurement Computing PCI-CTR05 5-channel counter/timer board at the direction of the MWT. The onset and duration of the stimulus were controlled through an Arduino custom-made-based interface.

## Optogenetic stimulation

For optogenetic experiments, light was delivered using a custom-made 16×16 LED panel. The arena was also illuminated from below with IR light. Light (alone or with air puff) was triggered at the 60th second and lasted for 30 seconds. The light intensity was measured as the irradiance (mW/cm²) using a PM16-130 photometer (THORLABS). Irradiance was measured at 12 points across the arena and then averaged. The light intensities used for optogenetic activation experiments (red light, 617 nm) were 0.1 mW/cm² for low, 0.2 mW/cm² for medium and 0.3 mW/cm² for high intensity. For optogenetic inactivation, 2 mW/cm² intensity was used for red (617 nm) light, and 0.9 mW/cm2 intensity was used for green (550 nm) light stimulation.

## Mechano- and thermo- nociceptive assays

Experiments were performed with staged third instar larvae as described for mechano-nociception[49] - and thermo-nociception assays[21,39]. The animals were staged for 6 h and allowed to develop for 4 days (96 h ± 3 h AEL). For mechano-nociception, larvae that crawled forward were stimulated on mid-abdominal segments (a3–5) with a 50 mN *von Frey* filament twice within 2 s. Each behavioral response was scored as nonnociceptive (no response, Stop, Stop and Turn) or nociceptive (C-shape bending, Rolling). Rolling and C-shape bending behavior are classified as nociceptive due to their absence in *TrpA1* mutant animals. Stopping, turning or no response were scored as nonnociceptive behaviors[50]. Each genotype was tested multiple times on different days, and the data from all trials were combined. Statistical significance was calculated using the chi² test.

For thermo-nociception via a local hot probe, a custom-built thermocouple device was used to keep the applied temperature constant at 46 °C. Staged and density-controlled third instar larvae (96 h ± 3 h AEL) were used, and all experiments were performed in a blinded fashion. Larvae that were crawling were touched with the hot probe on mid-abdominal segments (a4–6) until the execution of nociceptive Rolling (up to 10 s). The animals were videotaped, and their Rolling latencies were analyzed in a blinded fashion using ImageJ (NIH, Bethesda). Each genotype was tested multiple times on different days, and the data from all trials were combined. Statistical significance was calculated via Kruskal–Wallis ANOVA and pairwise comparisons with Dunn's *post hoc* test.

## Behavioral classification and analysis

Behaviors were assessed using an updated version of the custom-made machine learning algorithm described in Masson et al. 2020[19]. Behaviors were defined as mutually exclusive actions, i.e., for each time point, only one unique action was associated with the larva. The time series of the contours and the spines of individual larvae were obtained using Choreography (MWT package). From these time series, various features were computed, such as the center of the larva and velocities. All key features are presented in Masson et al., 2020. Behavioral classification consists of a hierarchical procedure that is trained separately on the basis of a limited amount of manually annotated data.

Here, we required an updated, more nuanced definition of behavior to better characterize the behavioral dynamics. This new definition is not mutually exclusive to the previous definition; rather, it refines it. We extended the hierarchy with an additional trainable layer to recast the two previous categories defined as Bends and Hunches into a more detailed behavioral category. What was previously categorized as "Bend" would now be classified as "Head Cast", "Static Bend" or "C-shape" (3 actions belonging to the general family "Bend"), and what was previously categorized as "Hunch" (Head-retraction) would now be classified as "Hunch", "Head-and-Tail" or "C-shape".(see the description of each behavior below). We used all the Bends and Hunches inferred by the first classification algorithm. With the beginning and end of actions being well identified, the last classification letter was designed and trained to split former behaviors into new subcategories. Actions were defined to be mutually exclusive and non-overlapping. Some actions, however, share similar dynamics (i.e. Head-and-Tail and Hunch, Head-and-Tail and C-shape and C-shape and Head Cast or Static Bend) when air puff and optogenetic stimulation were applied simultaneously.

## Action definition

The refined behavioral dictionnary is:

**Head Cast**. This is a subcategory of the previous Bend behavior. It is a dynamic bend where the head moves laterally from one side to the other. This behavior has been described in publications where larval taxis are described for chemotaxis or anemotaxis[28,29]. There are two subcategories within Head Cast: one for when the head moves strongly but the tail moves at the same speed as the center of mass (slower Head Cast) and another for when the tail moves rapidly (fast Head Cast).

**Static Bend**. This is a subcategory of the previous Bend behavior. It is a low-speed turning movement, with minimal head movement, and the angle between the segment between the center of mass and the head and the center of mass and the tail remains constant.

**Hunch**. This definition remains the same as previously described with an improved criterion for the straightness of the larva. It is a quick behavior, where the head of the larva retracts, decreasing its length between the center of mass and the head of the larva.

**C-shape**. This is a subcategory of Bend behavior when the larva takes the shape of the letter C, i.e., the head and the tail bend on the same side and remain relatively still.

**Head-and-Tail**. Simultaneous retraction of the head and the tail looking like a Hunch from both sides of the body of the larva. This behavior is classified as Hunch, Back or Run in the previous classifier because of the motion of the center of mass.

The updated definitions of behavior emerge naturally from directly plotting larval dynamical features. In Supplementary Fig. 15, we show the natural clustering of the behavioral categories based on the features used for the refined behavioral classification introduced.

**New annotated data.** We required new annotated data to ensure that the classifier matched the phenotype of the larva used in these experiments. The sets of C-shape, Hunch and Head-and-Tail were manually tagged from actions selected using the Masson et al. (2020) previous behavior classification pipeline. A few tags were used for the model.

Head Cast behavior was automatically tagged randomly from Bends before the stimulus. Static Bends were actions that occurred only during stimulation. The Static Bends was automatically tagged

with a threshold on the value of the velocity; if a Bend occurred during n time step, the motion velocity normalized by a length of 50% of the $n$ time step must be under $0.02\,s^{-2}$. We set a low threshold to avoid influencing the algorithm with erroneous annotations.

**New features.** To train the last layer of the pipeline, we combined features already evaluated with the previous layer of the classifier with new features. All features were averaged over the duration of the action; for each type of feature, we extracted the maximum and minimum values.

- The three velocities, the head velocity, the motion velocity and the tail velocity, were normalized by the length of the larva. The motion velocity is the velocity of the center of mass returned by the MWT software. Head-and-Tail are the terminal points of the spine. Averaging along the spine curve and its derivative, $S = \frac{1}{2}(3\langle cos^2\theta\rangle - 1)$, where $\cos\theta$ is the scalar product between normalized vectors associated with a segment of the spine and the direction of the larval body.
- The shape factor $\lambda = \frac{\lambda_1 - \lambda_2}{\lambda_1 + \lambda_2}$, where $\lambda_i$ represents the eigenvalue of the mean covariance matrix of movement, characterizes the shape of the larva and takes a value between 0 and 1.

The following features were also averaged over 5 time points before and 5 time points after the behavior.

The new features were as follows:

- The ratios between the length of the head-center of mass and the tail-center of mass $\frac{||\overrightarrow{HG}||}{||\overrightarrow{TG}||}$ with $H$, $T$ and $G$ are the coordinate points of the head, the tail and the center of the mass, respectively.
- Projection of the Head-and-Tail velocity on the spine of the larva. If we note the velocity vector of the head $HV_h$ with the H coordinate point of the head and $V_h$ the coordinate point at the end of the velocity vector, the projection point satisfies the basic relationship $||V_h - V'_h|| = \min||V_h - x||$ with $x \in \overrightarrow{r_{si}}$.
- The cosine of the angle between the vector of the head (tail) velocity and the first (last) segment of the larva, $\cos(\theta) = \frac{\overrightarrow{r_{s1}} \bullet \overrightarrow{v_{head}}}{||\overrightarrow{r_{s1}}||\,||\overrightarrow{v_{head}}||}$, where $\overrightarrow{v_{head}}$ is the vector velocity of the head and $\overrightarrow{r_{s1}}$ is the first vector of the spine.

All features were normalized by the length of the larva to ensure scale-free properties.

**New classifications.** We used a random forest to perform the classification and divided the process into two steps. The first layer separated the original Bends and Hunches into new Hunches, Static Bends and Head Casts. In the second layer, the new Hunches were further categorized into Hunches, Head-and-Tail and C-shape. The performance of the classification is illustrated in Fig. 2a. The GtACR1 data were analyzed separately using a second classifier, with tags specific to the GtACR1 data. This classifier distinguished between red light and green light conditions, as larval dynamics differed between these two conditions. Since the data pertained to a new genotype, the baseline features, such as speed and curvature, differed. Similar to the first classifier, the analysis process involved two stages.

We calculated the cumulative probabilities of the actions Crawl, Head Cast, Stop, Hunch, Back-up, Roll (as described in Masson et al. 2020) within different time windows after stimulus onset only in larvae that were tracked at the beginning of the stimulus.

**Statistical analysis**
Chi² tests were used for the statistical analysis of the behavioral probabilities. False rate discovery was used in the case of multiple testing with the Benjamini-Hochberg procedure[51]. When the Benjamini–Hochberg procedure was performed, the FDR was reported. The FDR in multiple comparisons was not used in cases where there were a few planned comparisons (e.g., Fig. 1 and Fig. 6). In those cases, the $p$ values were reported.

**Comparison of velocity distributions**
We compared some velocity distributions during the Head Cast and the Crawl to identify another type of response to the stimulus. We plotted the velocity distribution with a kernel density estimation (using Gaussian kernels). To assess whether two velocity distributions could originate from the same underlying distribution, we used the two-sample Kolmogorov–Smirnov test.

The two-sample Kolmogorov–Smirnov test was used to compare velocity distributions; it is a nonparametric hypothesis used to test whether two underlying one-dimensional probability distributions differ. The two distributions were derived from two different datasets. The Kolmogorov–Smirnov statistic is as follows: $D_{n,m} = sup_x(|F_n(x - G_m(x)|)$ where $F_n(x)$ and $G_m(x)$ are the empirical distribution functions of the first and second samples, respectively, and sup is the supremum function. The null hypothesis assumes that the data from the first sample and the second sample are from the same continuous distribution. The $p$ value is computed from the value of $D_{n,m}$.

**Calcium imaging**
We used both intact larvae and filet preparations (where the cuticle was left attached to the CNS). For intact larval imaging, third instar larvae were rinsed in water and mounted between a 2 cm circular coverslip and a custom-made device that delivers mechanical stimulation in low-melting-point 4%agarose (melted in phosphate-buffered saline), with the ventral side facing up. Larvae were gently squeezed in this position until the agar cooled so that the ventral nerve cord could be imaged through the cuticle.

Filet preparation was performed as previously described in Jovanic et al., 2016[17]. A precise dissection involved a longitudinal incision along the larva's midline, carefully removing all organs except nerve connections from the CNS to the cuticle (sensory organs and muscles). The cuticle was carefully stretched and secured at the corners for optimal exposure of the VNC. The dissection was performed in a calcium solution (39% NaCl, 1.8% KCl, 1.1% CaCl2, 2% MgCl2, 57% TES, and 61% sucrose) to safeguard neuronal function. Additionally, delicate nerve connections were handled with care to prevent damage.

Mechanical stimulation was generated by a waveform generator (Siglent sdg1032x) connected to a quick-mount extension actuator (Piezo Systems, Inc.), which was embedded in the Sylgard-coated recording chamber. The stimulation was set at 1,000 Hz, with an intensity of 1 to 20 V applied to the actuator. The amplitude of the acceleration produced by the actuator was measured thanks to with a triple-axis accelerometer (Sparkfun electronics ADXL313) connected to a RedBoard (Sparkfun electronics) and bound to the Sylgard surface with high vacuum grease. The acceleration was $1.14\,m.s^{-2}$ at 20 V and $0.61\,m.s^{-2}$ at 10 V. Mechanical stimulation was precisely triggered by the Leica SP8 software with the Leica "Live Data Mode" and to a trigger box branched to the scanning head of the microscope. A typical stimulation experiment consisted of 5 s of recording without stimulation, then 5 s of stimulation, and 5 s of recording in the absence of stimulation.

For optogenetic activation during in vivo imaging, larvae were mounted in the dark, with the least intensity of light possible in the room, to avoid nonspecific activation of the targeted neurons. The optogenetic stimulation of CsChrimson was achieved by a 617-nm wavelength LED (Thorlabs, M617F2), which was controlled by an LED driver (Thorlabs, LED1B) connected to the waveform generator and conveyed through a Ø 400 μm Core Patch Cable (Thorlabs) to the imaging field. Optogenetic stimulation was triggered at 50 Hz with 50% and 100% duty for 1 s, concomitantly with mechanical stimulation via the waveform generator. The irradiance was measured at the

level of the imaging field at 500 μW using a PM16-130 THORLABS photometer.

Imaging was performed using 2-photon scanning with a Leica SP8 microscope at 200 Hz, with a resolution of 512 × 190 pixels. At this resolution, the rate of acquisition was 1 frame/s, depending on the experiment. Some experiments were performed at 50 frames/s, with a resolution of 256×256.

A set of stimulations of different intensities was repeated the same number of times for each larva (3-8 times depending on the experiment). A resting interval of 60 s was used between each stimulation. The orders of the stimulation were randomized to exclude potential bias from a specific order of stimulation. The randomization was performed by associating one number with one specific stimulation and creating randomized suites of these numbers www.dcode.fr.

Neuronal processes were imaged in the VNC at the axonal level, and fluorescence intensity was measured by manually drawing a region of interest (ROI) in the relevant areas using custom Fiji macros. The data were further analyzed using customized MATLAB scripts. F0 was defined as the mean fluorescence in the ROI during baseline recording in the absence of a mechanical stimulus or optogenetic activation. $\Delta F/F0$ was defined at each time step t in the ROI as follows: $\Delta F/F0 = (F(t) - F0)/F0$. The means were calculated using the same number of repetitions per larva. Experiments showing activity before mechanical stimulation or optogenetic activation were removed from the analysis.

### Immunohistochemistry labeling
To determine the neurotransmitter identity of the neurons, immunolabeling was performed on split GAL4 lines, Gal4 lines crossed with UAS-myr::GFP flies, or LexA lines crossed with LexAop-myr::GFP. The VNC was dissected from third instar larvae and fixed with 4% PFA for 45 min at room temperature. After rinsing in PBS, ten minutes of permeabilization in PBS-T and two hours of blocking in 1% PBS-T-BSA, the CNS preparations were incubated at 4 °C (one to three nights) with the primary antibodies raised against neurotransmitters and GFP in PBS-T (mouse anti-CHAT (DSHB), 1/50; rabbit anti-GABA (SIGMA), 1/500; rabbit anti-DVGLUT (gift H. Aberle), 1/500; chicken anti-GFP (Invitrogen), 1/1000). Then, they were incubated at 4 °C (one to two nights) with fluorophore-coupled secondary antibodies in PBS-T raised against species of the primary antibodies(anti-mouse Alexa Fluor 647 (Jackson ImmunoResearch), 1/200; anti-rabbit Alexa Fluor 647 (Jackson ImmunoResearch), 1/200; anti-chicken Alexa Fluor 488 (Abcam), 1/1000). After rinsing, the preparations were mounted in antibleaching mounting medium (SlowFade Gold, Thermo Fisher S36939) under a cover slip. The confocal images were captured with a Leica SP8 confocal laser microscope. Alexa Fluor 488 was excited with a laser light of 488 nm, Cy3 with a laser light of 561 nm, and Alexa Fluor 647 with a wavelength of 633 nm.

### Optogenetic activation using SPARC
For the SPARC (Sparse Predictive Activity through Recombinase Competition) experiment[41], y[1] w[*]; P{y[+t7.7} w[+mC]=nSyb-IVS-phiC31} su(Hw)attP5/Cyotb;11A07-Gal4/Tm6TbSb was crossed with TI{20XUAS-SPARC2-I-Syn21-CsChrimson::tdTomato-3.1}CR-P40 and TI{20XUAS-SPARC2-S-Syn21-CsChrimson::tdTomato-3.1}CR-P40. Among the population of third-instar larvae, larvae were preselected depending on the phenotype exhibited in response to red light (escape or startle responses). The behavior of selected single larvae upon optogenetic activation was recorded with MWT (as above). For optogenetic activation, red light of 617 nm, as described, was used at 0.3 mW/cm² intensity. For selected larvae, dissection of the CNS was performed using immunochemistry experiments. Neurons expressing CsChrimson-TdT tomato were revealed using rabbit anti-DsRed (Clontech, 1/500) as the primary antibody and anti-rabbit-Cy3 (Jackson ImmunoResearch, 1/500) as the secondary antibody.

### EM reconstruction and connectivity analysis
EM reconstruction was performed using a complete CNS serial section transmission EM volume from a 6-hour-old *Canton S G1*c 3 c*w1118 [590S]* larva, with a resolution of 3.8 nm × 3.8 nm × 50 nm[18]. We used the web-based software CATMAID[52] to annotate the synapses onto and from A19c and reconstruct the 3D morphology of neurons presynaptic and postsynaptic to A19c. To do so, we used a previously described methodology[18,36]. The putative TDN was reconstructed up to recognition, a point at which we could confirm that the neuron morphology, including the position of the soma, dendrites and axon, matched the features of the neuron we observed in the light microscopy images. If at least one of the hemilateral pairs of neurons received at least 3 synapses from a particular neuron of interest (A19c, TDN, or their partners), we considered them strong downstream partners. For Basin neuron analysis (Supplementary Fig. 1), we considered neurons as not reconstructed up to recognition when the number of nodes was less than 1,500.

### Reporting summary
Further information on research design is available in the Nature Portfolio Reporting Summary linked to this article.

## Data availability
All data supporting the findings of this study are available within the paper, its Supplementary Information, as source data and/or within the repository: https://doi.org/10.5281/zenodo.13889569[53]. Source data are provided with this paper.

## Code availability
Scripts used in this paper are available at: https://doi.org/10.5281/zenodo.14192556.

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

## Acknowledgements

We thank the Cardona lab at the MRC LMB, UK, for hosting the EM volume in their CATMAID server which allowed us to reconstruct neurons and synapses. We thank Casey Schneider-Mizell, Laura Herren, Akira Fushiki, Eri Hasegawa, Ingrid Andrade, Aref Arzan Zarin, Avinash Khandelwal, Tara Guillorit and Claire Julliot De La Morandiere for their contribution to the EM reconstruction. We thank Alexandre Blanc, for his help with the Supplementary Fig. 15. We thank Ellie Heckscher for helpful discussions and comments throughout the project. We thank Daniel Vasiliauskas for comments and feedback on the manuscript. This work was supported by ANR PIA funding: ANR-20-IDEES-0002 (T.J) Agence Nationale de la Recherche (ANR-17-CE37-0019-01) (T.J.), ANR-NEUROMOD (ANR-22-CE37-0027) (T.J.), CNRS - UChicago collaboration grant (T.J.), Équipe FRM EQU202303016317 (T.J.), Tramway, ANR-17-CE23-0016 (J.B.M.), the inception Project PIA/ANR-16-CONV-0005,OG (J.B.M.), Investissement d'avenir programme under the management of ANR, ANR-19-P3iA-0001 (PRAIRIE 3IA Institute (J.B.M.), This work has

benefited from french government grants managed under the France 2030 program managed by the Agence Nationale de la Recherche (ANR-23-IAHU-0003: IHU reconnect & IHU ICE) (J.B.M.), the Deutsche Forschungsgemeinschaft DFG SO 1337/7-1 (P.S.) and the DFG Heisenberg program, SO 1337/6-1 (P.S.). This project has received funding from the European Union's Horizon 2020 research and innovation programme under the Marie Sklodowska-Curie grant agreement No 798050 (T.J. & J.B.M.). The funders had no role in study design, data collection and analysis, decision to publish, or preparation of the manuscript.

## Author contributions

M.L. behavioral experiments and analysis, EM reconstruction and analysis; Writing: results, original draft, figures C.B. behavioral classification and analysis, statistical analysis, writing: methods A.H. Calcium-imaging, SPARC, behavioral and immunohistochemistry experiments, figures. S.A SPARC and immunohistochemistry experiments, writing: methods, B.F. Ca-imaging experiments, N.D. behavioral experiments and analysis P.S. supervision, writing - edits and revisions J.B. Methodology, supervision, funding acquisition, writing- edits and revision T.J. Conceptualization, analysis, supervision, funding acquisition and project administration; writing: original draft, edits and revisions

## Competing interests

The authors declare no competing interests.
