## [Peer review File · Nature Communications]

REVIEWER COMMENTS

Reviewer #1 (Remarks to the Author):

This study identifies particular interneurons that mediate the competition between two dominant behavioral types in *Drosophila* larvae. Previously, Jovanic et al. (2016) demonstrated that activating Basin-1 alone results in hunching behavior, whereas simultaneous activation with Basin-2 leads to bending and suppresses hunching. The current paper furthers this work by pinpointing neurons downstream of the Basins, delineating their roles in behavior selection. Although these findings add valuable pieces to the neural circuitry puzzle, they do not represent a significant conceptual advance.

Additionally, the paper shows that varying the optogenetic activation intensity of R11A07 neurons, alongside differing air puff strengths, determines behavioral outcomes. While the influence of R11A07 activation levels presents new data, the underlying principle aligns with expectations set by prior research in similar studies. Moreover, it is unclear whether variations in defensive behavior are attributable to changes in stimulus context or intensity.

The manuscript offers exhaustive behavioral descriptions that, while detailed, contribute minimally to conceptual advances. Coupled with a lot of grammatical errors, this paper is difficult to read.

Other Major Concerns:

1. The redefinition and regrouping of established behaviors—such as C-shape bending and hunching, traditionally linked to nociceptive and mechanosensory circuits respectively—muddles the boundary between behavioral responses of nociceptive circuits and those of mechanosensory circuits.
2. Neuronal activity inhibition was conducted using TNT or Kir2.1, with expression spanning the developmental timeline. Given the study's emphasis on circuit analysis, mitigating these inhibitors' developmental impact is important. In key experiments, at least, the employment of transgenic lines designed for optogenetic inhibition would be advisable.

Minor Concerns:

1. Page4 line208. The authors says that "...Furthermore, optogenetic activation of early-ELs during air puff resulted in increased Hunch probability in response to air puff (Fig. 1H, Supplementary fig.

3A,B), showing that early-ELs promote Hunching, as do their presynaptic partners Basin-1.....” But Figure1H or Supplementary fig.3A,B do not seem to show increased Hunch probability. In figure 1H, there is no statistical difference, even with this large number of samples.

2. In the method section (Page17 Line746) says that 0.4 mW/cm² is high intensity for optogenetics stimulation. But the figure3 legend and several other places say 0.3 mW/cm² is high intensity.

3. If I understand it correctly, Figure3D and Figure2B2 are the same data. The authors should point out so in the manuscript. And, Sup Fig 9A and Sup Fig10A are exactly the same.

4. Fig5A and Sup Fig7G are the same pictures.

5. Page6 Line260, the author says “...when the light was delivered alone (Fig. 2B-D, Supplementary table 3)...”. There is no figure 2D.

6. Page8 Line364, The Fig4 is not relevant.

7. Several figure legends need to be corrected:

1) Sup Fig2 legend. There are no A4-5 or B4-5 panels in the figure but the figure legends include A4-5 and B4-5. There are many typos in the paragraph.

2) Figure2 legend, “Cs Crimson” should be “CsChrimson”

3) Figure5 legend, Label “A” is missing.

4) Figure7 legend, should be “R11A07-Gal4”, not “11A07-Gal4”.

5) Sup Fig4. The format for intensity is inconsistent; some have comas, some don't.

8. Figures issues:

1) In Sup Fig 2B. the label of y axis is wrong. should be 0.2/0.4/0.6/0.8 not 0,2/0,4/0,6/0,8. And Airpuff+light is missing the “ht” in B1 lower panel.

2) Figure1 G3. The X-axis label for “Static bend” is off.

3) Sup Fig9 C4, ethogram is missing.

4) Sup Fig7 E,F,G do not match the figure legends. The label in the figure should be 'e1,e2/f1,f2/g1,g2'.

9. There are many typo and formatting issues, I list a few here but there are more:

1) Page6 Line256 should be “Fewer” not “ewer”.

2) Page6 Line 278, a period is missing between “...activation” and “Medium-high....”.

3) Page8 Line367-370, extra space between number and “%”.

4) Page9 Line392, space missing in the end of the sentence.

5) Page17 Line758, “Chi²”, “2” should be superscripted.

Reviewer #2 (Remarks to the Author):

Lehman et al. extend previous studies by the Jovanic and Masson groups by expanding the known circuit in the *Drosophila* larva that transforms multisensory inputs (nociception and mechanosensation) to action selection of different behavioral outputs (start and escape responses). Making excellent use of the connectome, this study moves from the Basin cells (the focus of previous studies) to the next layer of circuitry (TDN, A19c, A08m, A08x) that differentially integrate sensory inputs and modulate different downstream behaviors. Improvements have been made in the automatic classification of behaviors from video microscopy (although it is hard to know whether this was a critical improvement or simply added greater detail). The many different experiments involving sensory inputs, optogenetic manipulations, and genetic perturbations appear internally consistent and well-connected to a qualitative model of sensory-to-motor transformation and some amount of lateral inhibition during action selection. Altogether, the experiments and analysis are thorough and persuasive. The study makes substantial steps into deeper layers of a fascinating circuit, and certainly deserves the broad audience of *Nature Communications*. I have no specific suggestions to improve the study, which is well done and thorough.

My major criticism is that the paper is hard to read. The text is dense and overly detailed. Much of the text is a verbal distillation of every aspect of the figures. Much of it is redundant. Not all of it is necessary. I strongly encourage the authors to simplify and shorten the Results. Hone in on main discoveries without documenting every detail. Focusing on the take-home message would make the paper more interesting. Details can be kept, but are better concentrated in figures and figure captions. I have no specific suggestions about how to do this rewriting, leaving this to the author's discretion. But my advice is for the authors to strive for clarity and concision. The paper makes sense, but its overly long and detailed presentation is exhausting and will reduce readership.

Reviewer #3 (Remarks to the Author):

In this manuscript, Lehman et al perform a rigorous analysis of a neuronal circuit in *Drosophila* larvae involved in defensive actions. They combine high-throughput, automated behavioral

analysis, optogenetics, and EM circuit reconstruction to carefully outline the involvement of a series of neurons in deciding the response of a larvae to threatening stimuli. The scientific approaches of the paper seem sound and the circuit reconstruction/analysis is extremely rigorous. However, the manuscript could potentially prove more interesting to a broad range of scientists if the writing and data presentation was simplified. Throughout the manuscript the contextualization of experiments, conclusions, and theorizing about behavior are extremely compelling and interesting, but the description of results and presentation of data is incredibly difficult to parse for non-experts. Often large sections of results are listed out one after another with many parenthetical breaks and little contextualization, requiring the reader to try and hold a large number of independent results in their mind at once before explaining how those results fit together in the larger picture. While the high-dimensionality of the data and complexity of circuit dissection makes the task of writing the results in an easy-to-understand fashion difficult, I think that a deliberate effort to make the manuscript simpler to read would greatly enhance its impact. This is of course not relevant to the scientific rigor of the study which is quite impressive. Minor notes are included below, largely concerning areas of the manuscript that could potentially be made more clear.

Minor Points:

- It would be very helpful to show a simplified diagram of the neuronal network and its associated behavioral outputs in earlier figures as you build the model, especially since it involves such a complex interplay of so many neurons coding so many different behaviors. Such diagrams are present in later figures and greatly aid the clarity of those experimental setups. If space is an issue, it could be the case that this kind of graphic would add more to readability than the graphs demonstrating the various probabilities of transitioning between behavioral states.
- The diagrams demonstrating the probabilities of transitioning between behavioral states might be clearer if the width of the line correlated with the strength of the connection.
- It would aid ease of reading for the legend showing what color indicates what behavior to be closer to the graph and in a more consistent location for graphs such as those in 1G. There also appears to be no such legend to explain the color meaning of 1 F1-2.
- It is unclear what is meant in the lines of the ethogram in figure 1a for individuals with white space before a color appears. Where the actions not categorizable during this time?
- There are so many parentheticals that it is often difficult to read sentences.
- The manuscript would benefit from copy editing, especially for simple typos.

Response to reviewers

We would like to thank the reviewers for spending time in thoroughly evaluating the paper. The revised version has taken into consideration their very useful comments. Please find in the following, in black the remarks from the reviewers, in blue our responses and between quotation marks portions of the text that have been added or modified in response to reviewers remarks.

Reviewer #1 (Remarks to the Author):

This study identifies particular interneurons that mediate the competition between two dominant behavioral types in *Drosophila* larvae. Previously, Jovanic et al. (2016) demonstrated that activating Basin-1 alone results in hunching behavior, whereas simultaneous activation with Basin-2 leads to bending and suppresses hunching. The current paper furthers this work by pinpointing neurons downstream of the Basins, delineating their roles in behavior selection. Although these findings add valuable pieces to the neural circuitry puzzle, they do not represent a significant conceptual advance. Additionally, the paper shows that varying the optogenetic activation intensity of R11A07 neurons, alongside differing air puff strengths, determines behavioral outcomes. While the influence of R11A07 activation levels presents new data, the underlying principle aligns with expectations set by prior research in similar studies.

Indeed, previous work made important advances on the understanding of neural circuit mechanisms of action selection in response to single sensory cue or in a multisensory context (Ohyama et al., 2015, Jovanic et al, 2016, Hu et al, 2017, Pan et al, 2023). To our knowledge, however, this is the first study that shows that nociceptive behaviors and responses to mechanosensory cues can mutually inhibit each other and sheds light on the neural circuit mechanisms underlying the selection of an appropriate defensive response in presence of two competing aversive cues. This mechanism is important for survival as when in the presence of two competing stimuli, like the situation that we mimic in our experiments with air puff and optogenetic activation of A19c, the appropriate behavioral response needs to be chosen and the choice will depend on the salience of each stimulus. This could constitute a general mechanism for implementing context-dependent modulation of behavioral decisions.

While previously published studies (Ohyama et al, 2015, and Pan et al, 2023) show that combining vibration with mechanical stimulation enhances rolling behavior and that gentle touch response can inhibit weak nociceptive responses, the findings in the present study shed light on the circuit elements underlying the competitive interactions between static and active defensive actions and the way these interactions are influenced by the sensory context which was not previously described. Specifically, it shows that mechanosensory and nociceptive behaviors can mutually inhibit each other, likely through winner-take-all type computation that depends on the level of activation of the neurons driving the different behaviors triggered by the different modalities.

We have made these points more explicit in the manuscript:

-In the abstract

-In the last paragraph of Introduction:

"Modulation of the behavioral output depends on the strength of activation of the second order interneurons relative to the strength of the mechanosensory context introduced by applying an air puff. Altogether, our work reveals that the level of activation of the identified neurons can gate specific responses depending on the context. These neurons thus contribute to the circuit computation for selecting appropriate defensive behavior, specifically in the presence of two aversive sensory cues: mechanosensory and nociceptive".

-We have added a new results section to better articulate this point: "Relative level of activation of A19c neurons determines the defensive behavior expressed" (I409-I432)

-In the discussion:

"Stronger air puff intensities can inhibit the Roll, while high optogenetic activation can inhibit the Hunch. This suggests that A19c and TDN neurons could gate the behavioral selection depending on contextual information towards more active or passive forms of avoidance behaviors. That competition is mutual: the nociceptive pathway elicits active and fast forms of escape and inhibits protective Hunching and Static Bending and conversely the mechanosensory pathway can inhibit escape behaviors in favor of the protective/startle-like actions."

"Additionally, they point to not unilateral gating, but rather mutual competition between mechanical and nociceptive responses. This could be important in case the animal is confronted with two types of dangers and it needs to decide which aversive stimulus to attend to as a priority, depending on the degree of each danger."

Moreover, it is unclear whether variations in defensive behavior are attributable to changes in stimulus context or intensity.

We thank the reviewer for pointing this out. It is true that in our experiments we vary both the sensory context (for example by adding an air puff) and its intensity. However, because we can compare the behavioral responses upon optogenetic activation with and without air puff, we can conclude that the presence of another stimulus (context) modulates the escape sequence. Furthermore, because we used two different intensities of air puff and observed intensity-dependent effects on escape sequences (the stronger the air puff the stronger the inhibition of escape behaviors), this suggests that this effect is dose-dependent. Of note, varying the intensity of air puff alone does not bias behavioral decisions but rather results in overall less responses at lower intensities of stimulation (Masson et al., 2020).

The experiments that combine different intensities of light (optogenetic activation) with different intensities of mechanical stimulation during behavior and calcium imaging reveal that the mechanical stimulation inhibits the activity of A19c neurons (and thus C-shape and Roll behaviors). Moreover, the sensory context (the presence of the mechanical stimulus) modulates the defensive behaviors in an intensity dependent manner (the calcium imaging data show that

the level of activation of the A19c neuron in the presence of mechanical stimulation decreases as a function of the stimulus intensity). Thus, in our view, the sensory context modulates the behavior by acting on the level of neuronal activation, likely through a neuron that receives mechanosensory input and can inhibit A19c. These competitive interactions between neurons will depend on their level of activation, and the level of neuronal activation will depend on the strength of the sensory input. Therefore the effects of the sensory context (and thus the behavioral outcome) will vary with its intensity as well as the intensity of the competing stimulus.

We have clarified this throughout the manuscript text and specifically in the paragraph : "Relative level of activation of A19c neurons determines the defensive behavior expressed", I404-424

The manuscript offers exhaustive behavioral descriptions that, while detailed, contribute minimally to conceptual advances. Coupled with a lot of grammatical errors, this paper is difficult to read.

We thank the reviewer for pointing this out. We have made significant changes throughout the manuscript to clarify behavioral descriptions and stress the important findings and conceptual advances. We have condensed the details in the description of behavioral experiments and moved those either to figure legends or method sections where appropriate. We have corrected typos and grammatical errors. The paragraphs in the manuscript with significant changes are highlighted in blue.

Other Major Concerns:

1. The redefinition and regrouping of established behaviors—such as C-shape bending and hunching, traditionally linked to nociceptive and mechanosensory circuits respectively—muddles the boundary between behavioral responses of nociceptive circuits and those of mechanosensory circuits.

We thank the reviewer for bringing up this point. We would like to clarify that the behavioral classification we provide in this study has been refined with respect to previously used methods (i.e. Masson et al, 2020). It does not rely on a redefinition of behavior and doesn't comprise categories resulting from regrouping previously defined behavioral categories.

The original definitions of behavior were designed to be used to analyze large amounts of screen data in defined stimulus conditions (i.e. air puff, Masson et al., 2020). Hence, in our study, as we focus on the context-dependent modulation of defensive behaviors, where we combine light (for optogenetic activation) with air puff stimulation, we updated the classification to include actions that typically do not occur upon air puff stimulation alone and to account for the difference in the larva dynamics when light is combined with air puff compared to air puff alone (or light alone).

For example, we define a previously uncharacterized behavior Head-and-Tail, that we discovered by inspecting the recordings of larva behavior. We observed an action that in addition to the retraction of the head, also comprised retractions of the tail (see Fig S4.A). Given

that our classification method until this paper detected six mutually exclusive categories of behaviors: Hunch, Bend, Stop, Back-Up Crawl and Roll (see Masson et al, 2020), the new actions were originally classified as one of these 6 categories. Since Head-and-Tail is the most similar to Hunch, the original classifier would classify it as a Hunch. Similarly we have observed C-shape in some experiments. Those were classified as either Bend (as they are most similar to Bends) and in some cases also were classified as Hunches as prior we identified these actions in the optogenetics and air puff and optogenetics combined experiment, all rapid retraction of the head during air puff responses were associated with the Hunch state regardless of the state of motion of the tail or if the larva took the slight curved shape (which does not happen during air puff stimulation alone). In addition, ongoing work in the lab had been looking at static versus active actions and visual inspection of behaviors categories, revealed that within the Bend category in addition to Head Casts that were previously described during foraging and navigation, there is category of bent behavior that didn't involve seeping the head on one of both sides, but instead a bend of various amplitudes that remained static. Also, different types of manipulations (internal state, neuron inactivation etc.) affect Head Casting and Static Bending differently. We therefore in the expanded classification method also replaced Bend with two behavioral categories (Head Cast and Static Bend). The existence of these actions, or at least the insurance that they have different representation within the CNS of the larva, is proved by the ability to control their relative balance by neural activation or control of the sensory environment or internal state.

To summarize, we didn't regroup any behaviors in terms of definitions, we subjected two categories identified by the 6-category classifier i.e. Hunch, Bend to a new round of reclassification so they could further be accurately classified into one of the following categories: Hunch, Head-and-Tail, Head Casting, Static Bend, C-shape. What was previously categorized as "Bend" would now be either classified as "Head Cast", "Static Bend" or "C-shape" (three actions belonging to the general family "Bend") and what was previously categorized as "Hunch" (a retraction of the head) would now be classified as either "Hunch", "Head-and-Tail" or "C-shape". The other actions: Stop, Back-up, Crawl were classified as previously described (Masson et al, 2020).

Our previous detailed description of the new classification that addressed both the technical procedure and the biological relevance of the additionally classified action may have been misleading. We have therefore clarified this in the manuscript and moved most of the technical details to the Methods section.

The paragraph in the main text now reads:

"To quantify the probability of the different actions in the escape sequence in different conditions of stimulation, we expanded the machine learning-based behavioral classification algorithms¹⁹ to also detect Head-and-Tail and C-shape actions (Fig. 2A). The new classification was computed using an extra layer of Random Forest on top of the previous classifier adding new sets of features and trained on newly annotated data (see detailed description in the Method section)."

And the paragraphs in the methods section:

"Behaviors were inferred using an updated version of the custom-made machine learning algorithm described in ⁵. Behaviors were defined as mutually exclusive actions, i.e. to each time point only a unique action is associated with the larva....

Here, we required an updated, more subtle, definition of behavior to better characterize the behavioral dynamics. This new definition is not mutually exclusive to the previous definition rather it refines it. We extended the hierarchy with an additional trainable layer to recast the two previous categories defined as Bends and Hunches into a more detailed behavioral category. What was previously categorized as "Bend" would now be either classified as "Head Cast", "Static Bend" or "C-shape" (3 actions belonging to the general family "Bend") and what was previously categorized as "Hunch" (Head-retraction) would now be classified as either "Hunch", "Head-and-Tail" or "C-shape" (see the description of each behavior below).

Actions were defined to be mutually exclusive and non-overlapping. Some actions however share similar dynamics (i.e. Head-and-Tail and Hunch, Head-and-Tail and C-shape and C-shape and Head Cast or Static Bend) when air puff and optogenetic stimulation when applied simultaneously."

In addition, although it is not intended as a proof of behavior existence, we show that simple features associated with larval motion lead to natural separation of the actions described in the paper (Supplementary figure 15).

2. Neuronal activity inhibition was conducted using TNT or Kir2.1, with expression spanning the developmental timeline. Given the study's emphasis on circuit analysis, mitigating these inhibitors' developmental impact is important. In key experiments, at least, the employment of transgenic lines designed for optogenetic inhibition would be advisable.

We have performed optogenetic inactivation experiments using GtACR1 in several relevant neurons. The early-born EL optogenetic inactivation gave a similar phenotype as TNT inactivation (less Hunching, see Figure S3E). Driving R11A07 with GtACR1 resulted in significantly less Head-Casting and surprisingly more Stopping and Backing up, while no Hunches could be detected when green light was used for stimulation.

While the difference in the phenotype (more Hunching in TNT inactivation). could come from the different types of manipulations (one that spans the whole life of the larva and one that is transient during the stimulation), we also thought to examine whether the presence of green light used for GtACR optogenetic inactivation could provide a different context compared to TNT experiments (where no additional external stimulation is necessary, except for the air puff) or red light that we used during optogenetic activation experiments. We therefore performed experiments where we used red light at high intensities that was also shown to be able to suppress spiking in neurons in which GtACR1 was expressed (Jonaitis et al, 2024). We found that in these conditions upon inactivation the larvae perform more Hunches similarly to the TNT inactivation experiments (Figure S2B,C).

The Stopping or immobilization phenotype upon GtACR1 inactivation with green light could be due to stronger light startle response. Indeed. Head Casting is indeed increased in controls upon applying green light compared to when for example red light was used. This

stronger Head Casting in response to light could impact the behavioral responses of larvae when green light is used for GtACR inactivation. Another possible explanation for the difference in phenotype with green and red light is a stronger level of hyperpolarization (and subsequent depolarization/activation if conductance for chloride is too high) could be the reason for a complete immobilization of a large proportion of larvae and the lack of Hunches upon air-puff onset.

We have included these data in Figures S2 and S3 and described the results in the results section of the manuscript (I185-I190, I213-I215).

Finally, we have performed the GtACR1 inactivation experiments for Basin-2 and Basin-4 neurons. We have previously found that these neurons inhibit Hunching as their inactivation using TNT resulted in more Hunching (Jovanic et al, 2016). The GtACR1 inactivation experiments showed similar phenotypes to those that were previously observed with TNT inactivation and confirmed that these neurons inhibit Hunching. We have included these data in Fig. S8.

Minor Concerns:

1. Page 4 line 208. The authors say that "...Furthermore, optogenetic activation of early-ELs during air puff resulted in increased Hunch probability in response to air puff (Fig. 1H, Supplementary fig. 3A,B), showing that early-ELs promote Hunching, as do their presynaptic partners Basin-1...." But Figure 1H or Supplementary fig. 3A,B do not seem to show increased Hunch probability. In figure 1H, there is no statistical difference, even with this large number of samples.

We thank the reviewer for pointing this out. We have toned down this statement (I215-218).

2. In the method section (Page 17 Line 746) says that 0.4 mW/cm² is high intensity for optogenetics stimulation. But the figure 3 legend and several other places say 0.3 mW/cm² is high intensity.

We thank the reviewer for noticing this. We have made a typo. The correct value for high intensity is 0.3 mW/cm². We have corrected this.

3. If I understand it correctly, Figure 3D and Figure 2B2 are the same data. The authors should point out so in the manuscript. And, Sup Fig 9A and Sup Fig 10A are exactly the same.

We thank the reviewer for pointing this out. Indeed the data in Figures 3D1 and 2B2 are the same. In Figure 3D1 they are shown again as they are used to compare the behavioral responses in the light alone condition with light + medium and strong air-puff conditions (shown in Fig D2 and Fig. 3D3). We have clarified this in the figure legend.

Fig. 9A and Fig. 10A represents the control that is the same for Basin-2 and All basins activation experiments, and are used in each image for comparison purposes. We have made this explicit in the figure legend.

4. Fig5A and Sup Fig7G are the same pictures.

Indeed these two images are the same. In Fig. 5A shows the expression profile of R11A07, while in Sup Fig 7 G it is used to identify the A19c and TDN neurons in different segments of the VNC that were labeled for vGlut, Chat and GABA. We have noted this in the Sup Fig. 7 legend

5. Page6 Line260, the author says "...when the light was delivered alone (Fig. 2B-D, Supplementary table 3)..." . There is no figure 2D.

Thank you for noticing this. We have corrected this

6. Page8 Line364, The Fig4 is not relevant.

We have corrected this

7. Several figure legends need to be corrected:

- 1) Sup Fig2 legend. There are no A4-5 or B4-5 panels in the figure but the figure legends include A4-5 and B4-5. There are many typos in the paragraph.
- 2) Figure2 legend, "Cs Crimson" should be "CsChrimson"
- 3) Figure5 legend, Label "A" is missing.
- 4) Figure7 legend, should be "R11A07-Gal4", not "11A07-Gal4".
- 5) Sup Fig4. The format for intensity is inconsistent; some have comas, some don't.

We thank the reviewer for noticing this. We have made those corrections in the figure legends

8. Figures issues:

- 1) In Sup Fig 2B. the label of y axis is wrong. should be 0.2/0.4/0.6/0.8 not 0,2/0,4/0,6/0,8. And Airpuff+light is missing the "ht" in B1 lower panel.
- 2) Figure1 G3. The X-axis label for "Static bend" is off.
- 3) Sup Fig9 C4, ethogram is missing.
- 4) Sup Fig7 E,F,G do not match the figure legends. The label in the figure should be 'e1,e2/f1,f2/g1,g2'.

We have corrected these issues

9. There are many typo and formatting issues, I list a few here but there are more:

- 1) Page6 Line256 should be "Fewer" not "ewer".
- 2) Page6 Line 278, a period is missing between "...activation" and "Medium-high....".
- 3) Page8 Line367-370, extra space between number and "%".
- 4) Page9 Line392, space missing in the end of the sentence.

5) Page17 Line758, “Chi²”, “2” should be superscripted.

We have corrected these issues and other typos and formatting issues

Reviewer #2 (Remarks to the Author):

Lehman et al. extend previous studies by the Jovanic and Masson groups by expanding the known circuit in the *Drosophila* larva that transforms multisensory inputs (nociception and mechanosensation) to action selection of different behavioral outputs (start and escape responses). Making excellent use of the connectome, this study moves from the Basin cells (the focus of previous studies) to the next layer of circuitry (TDN, A19c, A08m, A08x) that differentially integrate sensory inputs and modulate different downstream behaviors. Improvements have been made in the automatic classification of behaviors from video microscopy (although it is hard to know whether this was a critical improvement or simply added greater detail). The many different experiments involving sensory inputs, optogenetic manipulations, and genetic perturbations appear internally consistent and well-connected to a qualitative model of sensory-to-motor transformation and some amount of lateral inhibition during action selection. Altogether, the experiments and analysis are thorough and persuasive. The study makes substantial steps into deeper layers of a fascinating circuit, and certainly deserves the broad audience of Nature Communications. I have no specific suggestions to improve the study, which is well done and thorough.

My major criticism is that the paper is hard to read. The text is dense and overly detailed. Much of the text is a verbal distillation of every aspect of the figures. Much of it is redundant. Not all of it is necessary. I strongly encourage the authors to simplify and shorten the Results. Hone in on main discoveries without documenting every detail. Focusing on the take-home message would make the paper more interesting. Details can be kept, but are better concentrated in figures and figure captions. I have no specific suggestions about how to do this rewriting, leaving this to the author's discretion. But my advice is for the authors to strive for clarity and concision. The paper makes sense, but its overly long and detailed presentation is exhausting and will reduce readership.

We thank the reviewer for their positive comments and suggestions. We have made significant changes in the manuscript text. We have especially made an effort to simplify and clarify the description of the results, to make it easier to comprehend. We have moved parts of technical details and detailed descriptions of figures either to figure legends or method sections where appropriate. All significant changes are highlighted in blue throughout the text and figure legends.

Reviewer #3 (Remarks to the Author):

In this manuscript, Lehman et al perform a rigorous analysis of a neuronal circuit in *Drosophila* larvae involved in defensive actions. They combine high-throughput, automated behavioral analysis, optogenetics, and EM circuit reconstruction to carefully outline the involvement of a series of neurons in deciding the response of a larva to threatening stimuli. The scientific approaches of the paper seem sound and the circuit reconstruction/analysis is extremely rigorous. However, the manuscript could potentially prove more interesting to a broad range of scientists if the writing and data presentation was simplified. Throughout the manuscript the contextualization of experiments, conclusions, and theorizing about behavior are extremely compelling and interesting, but the description of results and presentation of data is incredibly difficult to parse for non-experts. Often large sections of results are listed out one after another with many parenthetical breaks and little contextualization, requiring the reader to try and hold a large number of independent results in their mind at once before explaining how those results fit together in the larger picture. While the high-dimensionality of the data and complexity of circuit dissection makes the task of writing the results in an easy-to-understand fashion difficult, I think that a deliberate effort to make the manuscript simpler to read would greatly enhance its impact. This is of course not relevant to the scientific rigor of the study which is quite impressive. Minor notes are included below, largely concerning areas of the manuscript that could potentially be made more clear.

We thank the reviewer for their positive comments and suggestions. During the revisions, we have made significant edits in the manuscript to make it easier to digest. We have particularly made an effort to simplify the description of the results by removing less relevant details (and where appropriate added those in the methods section and/or figure legends). All changes in the text of the manuscript are highlighted in blue.

Minor Points:

- It would be very helpful to show a simplified diagram of the neuronal network and its associated behavioral outputs in earlier figures as you build the model, especially since it involves such a complex interplay of so many neurons coding so many different behaviors. Such diagrams are present in later figures and greatly aid the clarity of those experimental setups. If space is an issue, it could be the case that this kind of graphic would add more to readability than the graphs demonstrating the various probabilities of transitioning between behavioral states.

We thank the reviewer for this suggestion. We have added the simplified diagram of the network in Figures 4C and Figure 9B.

- The diagrams demonstrating the probabilities of transitioning between behavioral states might be clearer if the width of the line correlated with the strength of the connection.

We thank the reviewer for this suggestion. We have made these changes so that the percentage of transition is proportional to the thickness of the transition arrows in all main and

supplementary figures with transition probabilities (Figure 1A, Figure 2 C1-C4, Supplementary Figures 5A3, B3, C3).

- It would aid ease of reading for the legend showing what color indicates what behavior to be closer to the graph and in a more consistent location for graphs such as those in 1G. There also appears to be no such legend to explain the color meaning of 1 F1-2.

We have made those changes (Figures 1F,G,H)

- It is unclear what is meant in the lines of the ethogram in figure 1a for individuals with white space before a color appears. Where the actions not categorizable during this time?

The larvae were not tracked during this time. In this ethogram all larvae that were present (even partially) during the first five seconds of stimulation are shown. We display on this ethogram all larvae present between 59.8 and 61 seconds for at least one time step. We have clarified that in the figure legend.

- There are so many parentheticals that it is often difficult to read sentences.
- The manuscript would benefit from copy editing, especially for simple typos.

We have many changes in the writing throughout the manuscripts that simplify and clarify the sentences. We have also copy-edited the text.

References

- Jovanic, T. *et al.* Competitive Disinhibition Mediates Behavioral Choice and Sequences in *Drosophila*. *Cell* 167, 858-870.e19 (2016).
- Ohyama, T. *et al.* A multilevel multimodal circuit enhances action selection in *Drosophila*. *Nature* 520, 633–639 (2015).
- Pan, G. *et al.* Cross-modal modulation gates nociceptive inputs in *Drosophila*. *Curr. Biol.* 0, (2023).
- Hu, C. *et al.* Sensory integration and neuromodulatory feedback facilitate *Drosophila* mechanonociceptive behavior. *Nat. Publ. Group* 20, 1085–1095 (2017).
- Masson, J.-B. *et al.* Identifying neural substrates of competitive interactions and sequence transitions during mechanosensory responses in *Drosophila*. *Plos Genet.* 16, e1008589 (2020).
- Jonaitis, J. *et al.* Steering From the Rear: Coordination of Central Pattern Generators Underlying Navigation by Ascending Interneurons. 2024.06.17.598162 Preprint at <https://doi.org/10.1101/2024.06.17.598162> (2024).

REVIEWERS' COMMENTS

Reviewer #1 (Remarks to the Author):

The revised manuscript is much improved. The new data strength the paper. The paper is also more readable than the previous version. I support its publication.

Reviewer #2 (Remarks to the Author):

I largely liked the paper before, I just found it was very dense, overly long, and hard to read. The presentation has improved. It is still enormously dense, requiring patience and endurance. I have never been able to get through Thomas Pynchon. This is about as hard. That said, I think the science is good, and the authors have faithfully attended to suggestions.

Reviewer #2 (Remarks on code availability):

N/A

Reviewer #3 (Remarks to the Author):

The work is still rigorous and I have no recommendations for any additional experiments. The editing of the manuscript has also made the paper much easier to read with a great deal of the extraneous detail removed. However, the paper is perhaps in even greater need of copy editing than before. There are many grammatical mistakes throughout, typos, unpaired parentheses, misused commas, and other errors. While I think the paper could be made even more compelling to the general reader than it is in its current, improved state, I think it would be reasonable to publish it as is as long as all of the errors in the paper are corrected.

Reviewer #3 (Remarks on code availability):

Beyond my expertise

RESPONSE TO REVIEWERS' COMMENTS

We would like to thank the reviewers for spending time in thoroughly evaluating the revised manuscript and were satisfied with the changes that were made to address the concerns that were raised. The final version has taken into consideration the comments about the remaining typos and language errors and we have thus sent the manuscript to a language editing service to make sure that any mistakes that we have missed would be corrected.

Reviewer #1 (Remarks to the Author):

The revised manuscript is much improved. The new data strengthens the paper. The paper is also more readable than the previous version. I support its publication.

Reviewer #2 (Remarks to the Author):

I largely liked the paper before, I just found it was very dense, overly long, and hard to read. The presentation has improved. It is still enormously dense, requiring patience and endurance. I have never been able to get through Thomas Pynchon. This is about as hard. That said, I think the science is good, and the authors have faithfully attended to suggestions.

Reviewer #2 (Remarks on code availability):

N/A

Reviewer #3 (Remarks to the Author):

The work is still rigorous and I have no recommendations for any additional experiments. The editing of the manuscript has also made the paper much easier to read with a great deal of the extraneous detail removed. However, the paper is perhaps in even greater need of copy editing than before. There are many grammatical mistakes throughout, typos, unpaired parentheses, misused commas, and other errors. While I think the paper could be made even more compelling to the general reader than it is in its current, improved state, I think it would be reasonable to publish it as is as long as all of the errors in the paper are corrected.

Reviewer #3 (Remarks on code availability):

Beyond my expertise